



# Separating snow and ice melt using water stable isotopes and glacio-hydrological modelling: towards improving the application of isotope analyses in highly glacierized catchments

Tom Müller[1,2], Mauro Fischer[2,3], Stuart N. Lane[1], and Bettina Schaefli[1,2,3]

[1]Institute of Earth Surface Dynamics, University of Lausanne, Lausanne, Switzerland
[2]Institute of Geography, University of Bern, Bern, Switzerland
[3]Oeschger Centre for Climate Change Research, University of Bern, Bern, Switzerland

**Correspondence:** Tom Müller (tom.muller1@gmail.com)

**Abstract.**

Glacio-hydrological models are widely used for estimating current and future streamflow across spatial scales, utilizing various data sources, notably streamflow and snow/ice observations. However, modeling highly glacierized catchments poses challenges due to data scarcity and complex spatio-temporal meteorological conditions, leading to input data uncertainty and

potential misestimation of snow and ice melt proportions. Some studies propose using water stable isotopes to estimate water shares of rain, snow, and ice in streamflow, yet the choice of isotopic composition of these water sources significantly impacts results. This study presents a combined isotopic and glacio-hydrological model to determine seasonal shares of snow and ice melt in streamflow for the Otemma catchment in the Swiss Alps. The model leverages available meteorological station data (air temperature, precipitation, and radiation), ice mass balance data and snow cover maps to model and automatically calibrate the

catchment-scale snow and ice mass balances. The isotopic module, building on prior work by Ala-aho et al. (2017), estimates seasonal isotopic compositions of precipitation, snow, and ice. The runoff generation and transfer model relies on a combined routing and reservoir approach and is calibrated based on measured streamflow and isotopic data.

Results reveal challenges in distinguishing snow and ice melt isotopic values in summer, rendering a reliable separation between the two sources difficult. The modelling of catchment-wide snow melt isotopic composition proves challenging due to

uncertainties in precipitation lapse rate, mass exchanges during rain-on-snow events, and snow fractionation. The study delves into these processes, their impact on model results, and suggests guidelines for future models. It concludes that water stable isotopes alone cannot reliably separate snow and ice melt shares for temperate alpine glaciers. However, combining isotopes with glacio-hydrological modeling enhances hydrologic parameter identifiability, in particular those related to runoff transfer to the stream, and improves mass balance estimations.

**Keywords.** combined isotopic and glacio-hydrological model, snow hydrology, snow and ice melt, water isotopes, highly glacierized catchments





## 1 Introduction

Highly glacierized catchments are rapidly changing due to climate change and subsequent glacier retreat (Huss and Hock, 2018; Zekollari et al., 2019). Reduced ice melt contribution, combined with more liquid precipitation and earlier snow melt, will significantly affect water resource availability (Berghuijs et al., 2014; Beniston et al., 2018). These changes will have a serious impact on downstream ecosystems (Milner et al., 2017), water usage for irrigation (Viviroli et al., 2020; Shokory et al., 2023), hydropower (Schaefli et al., 2019), or other domestic water uses (Immerzeel et al., 2020), both in densely populated lowlands (Pritchard, 2019; Biemans et al., 2019) or in small communities at high elevation (Buytaert et al., 2017).

In this context, glacio-hydrological models have been developed to assess current and future streamflow changes (e.g., Farinotti et al., 2012; Muelchi et al., 2022). Such models are usually calibrated based on different observational data sets, the most common being streamflow, followed by ice and snow related data (van Tiel et al., 2020). The resulting streamflow projection corresponds to the statistically best possible representation of the calibration or evaluation datasets, but the underlying physical streamflow generation processes remain simplified (Schaefli et al., 2011). In addition, field observations in high mountain catchments, where access is difficult and environmental conditions are harsh, are usually sparse, while the measured parameters may be subject to large spatio-temporal variations. The main drivers of annual glacier mass balance variability and glacier evolution are well documented (e.g., Huss et al., 2021). However, the lack of direct observations of processes below the ground or ice surface leads to model simplifications, such as for instance the effect of debris transport and supraglacial debris cover (Jouvet et al., 2011; Ayala et al., 2016), lateral subsurface flow (Carroll et al., 2019), permafrost melt (Rogger et al., 2017) or superficial and deep groundwater recharge and exfiltration (Hood and Hayashi, 2015; Penna et al., 2017). As a result, models may (i) wrongly represent the competing amounts of modelled snow and ice accumulation and melt, (ii) compensate runoff errors in the glacierized and non-glacierized parts of the catchment (van Tiel et al., 2020) or (iii) overlook or oversimplify processes which may become more dominant in the future such as for e.g. water storage and delayed water release (Somers and McKenzie, 2020). In addition, the associated uncertainties in the predicted catchment-scale streamflow remains difficult to evaluate.

The use of other types of glacio-hydrological information (in addition to streamflow and snow or ice melt data), such as the share of different water sources in streamflow (rainfall, snow melt, ice melt, older groundwater), can provide additional constraints to improve the representation of internal processes in glacio-hydrological models. In snow-dominated, non-glacierized catchments, stable isotopes of water have been used to estimate the shares of snow and rainfall in streams or groundwater (Ala-aho et al., 2017; Carroll et al., 2018; Beria et al., 2020). Indeed, the water isotopic composition of snow is more depleted in heavy isotopes ($^{18}$O and $^2$H) versus light isotopes ($^{16}$O and $^1$H) compared to rainfall. This phenomenon called fractionation occurs because, as air temperature decreases, lower cloud condensation temperature leads to more vapor condensation and a preferential loss of heavy isotopes from the air masses to the liquid phase. This results in an air mass becoming gradually more depleted in heavy isotopes (more negative $\delta^2$H ) as condensation occurs.

The isotopic signature of water is expressed as the ratio of heavy over light isotopes compared to the Vienna Standard Mean Ocean Water (VSMOW) (see Eq. 1), and has usually a negative value indicating the degree of depletion in heavy isotopes





(Beria et al., 2018). In hydrological studies, the water isotopic signature appears as an ideal tracer, as water conserves the isotopic composition from its source (rain, snow or ice melt) along hydrological flowpaths. Mixing models have been used to estimate the share of the different water sources in the stream based on the isotopic composition of the stream water and of the possible sources of water that compose it (the so-called "end-members").

$$\delta^2 H = \frac{\left(\frac{^2\mathrm{H}}{^1\mathrm{H}}\right)_{\mathrm{sample}} - \left(\frac{^2\mathrm{H}}{^1\mathrm{H}}\right)_{\mathrm{VSMOW}}}{\left(\frac{^2\mathrm{H}}{^1\mathrm{H}}\right)_{\mathrm{VSMOW}}} \tag{1}$$

In glacierized catchments, only a few studies have attempted to use water isotopes to provide an estimate of the shares of snow and ice melt at the time of sampling (e.g., Engel et al., 2016; Penna et al., 2017). Despite the emergence of encouraging research, many challenges regarding the use of water isotopes for snow and ice dominated catchments have been discussed. Spatial variations in the ice and snow melt isotopic signature may be large and a limited number of samples may lead to
biases (Engel et al., 2016; Schmieder et al., 2018; Zuecco et al., 2019). The snowpack isotopic signature tends to become more enriched in heavy isotopes over the melting season due to snow fractionation (Beria et al., 2018). The temporal evolution of the isotopic signal of snow appears therefore difficult to characterize and the choices of the selected "end-member" values lead to a significant trade-off between snow and ice melt contributions (Penna et al., 2017). Finally, snow melt isotopic signature may become similar to ice melt during the melting season, rendering their separation difficult.

The complex snow isotopic processes have been studied experimentally (e.g., Taylor et al., 2001; Carroll et al., 2022a; Sprenger et al., 2023), and equations characterizing the isotopic enrichment of snow have been proposed (Feng et al., 2002; Ala-aho et al., 2017). In addition, a limited number of studies have successfully used water isotopes for glacio-hydrological model evaluation (Hindshaw et al., 2011) or calibration (He et al., 2019; Nan et al., 2022). These studies show promising results, with a reduction of the uncertainties in both the estimation of model parameters and the shares of the water sources
when water isotopes are used for calibration.

In this study, we aim to further explore the use of water isotopes in glacierized catchments and to ultimately assess how water isotopes can be used to better approximate the shares of snow and ice melt. This requires in particular a detailed spatio-temporal representation of the snow melt isotopic composition, a characterization of the ice melt isotopic composition and an assessment of the contribution of rain events at the outlet.

For this purpose, we adapted the parsimonious model proposed by Ala-aho et al. (2017) to the case of a heavily glacierized catchment in the Swiss Alps and propose a combined isotopic and glacio-hydrological model which precisely estimate the temporal isotopic enrichment of snow melt based on physical processes. The main processes included in the model are (i) rain-on-snow refreezing in the snowpack, (ii) snow sublimation where isotopically lighter water tends to leave the snowpack, resulting in an isotopically heavier snowpack (less negative $\delta^2$H , proportionally enriched in heavy isotopes) and (iii) snow
melt from the snowpack resulting also in a isotopically heavier snowpack.

Since data collection is challenging in high mountain environments, we first develop a combined mass balance and isotopic model based on meteorological and glacier mass balance observations and a reasonable number of isotopic samples, which can be easily applied to other catchments. Measured discharge at the glacier portal is used in a second step to simulate water




transfer to the outlet (via hillslope, snowpack and glacier routing), which allows us to assess the impact of delayed water
transport on the isotopic composition of the stream. Finally, we compare results obtained from an end-member mixing model
based on isotopes only with the results from the glacio-hydrological model and propose guidelines for a robust definition of
isotopic end-members in highly glacierized catchments.

## 2   Study site and experimental methods

### 2.1   The Otemma glacier

The Otemma glacier is located in the southwestern Swiss Alps (45°57'00"N, 7°26'51"E) and is amongst the 15 largest Swiss
glaciers (Linsbauer et al., 2021). The glacier is characterized by a long and flat main lobe flowing in northeast-southwest
direction, and several steeper tributary glaciers to the southeast (Fig. 1). Due to its limited area at high elevation and large
proportion of the remaining ice volume within the ablation area, Otemma glacier has shown rapid retreat (2500 m or about
40 m per year since the 1970s, GLAMOS (1881-2020)) and comparatively large volume and mass loss (Fischer et al., 2015)
in recent decades. Most of the glacier is projected to completely melt already by 2060 under current climate change, and
to completely disappear by the end of the century (Gabbi et al., 2012). Two medial moraines deliver supra- and englacial
sediments to the glacier terminus, especially in its more shaded southern part, where the terminus gradually becomes heavily
debris-covered. Except for this area, the glacier mostly consists of relatively clean ice, and has a relative debris cover of about
12 % (estimated from Linsbauer et al., 2021).

The catchment boundary was defined about 100 m below the location of the main glacier portal in 2019, where a gauging
station was installed (Fig. 1). It has an area of 20.8 km$^2$, a mean elevation of 3080 m a.s.l. (2470 to 3730 m a.s.l.) and a
glacier coverage of 56 % in 2019 (adapted from Linsbauer et al., 2021). The underlying bedrock consists of orthogneiss and
metagranodiorites (Burri et al., 1999), overlain by coarse superficial sediment deposits with limited vegetation development.

### 2.2   Meteorological observations

In September 2019, a weather station was installed 50 m from the glacier terminus, at an elevation of 2450 m a.s.l. (Fig. 1),
and continuously recorded meteorological data with a resolution of 5 minutes until October 2021 with a 5 minutes resolution.
Liquid precipitation was measured with a Davis tipping rain gauge, air temperature, relative humidity and air pressure with
a Decagon VP-4, and incoming shortwave radiation with a SP-110-SS from Apogee Instruments. Solid winter precipitation
was extrapolated to the analysed catchment from two nearby MeteoSwiss automatic weather stations: from Otemma (∼5 km
downvalley, at 2357 m a.s.l.) for the winter 2019/2020, and from Arolla (∼10 km northeast, at 2005 m a.s.l.) for the winter
2020/2021. Wind speed was only available for the MeteoSwiss weather station of Grand St-Bernard (2472 m a.s.l.) about 20
km west of the Otemma glacier. All meteorological data have already been published (Müller, 2022).





**Figure 1.** The Otemma glacierized catchment with the locations of i) the gauging station close to the glacier portal, ii) the weather station, iii) 10 ablation stakes used for summer surface ice melt measurements and iv) two snow pits for average end-of-winter (28.05.2021) snow density estimation and isotope sampling as well as 92 end-of-winter snow accumulation measurements (28.05.2021, in snow water equivalent (SWE)). A third snow pit used for isotopic sampling is also shown in the glacier forefield. The black grid represents the perimeter and cell size for mass balance and snow isotopic composition modelling. The small inset indicates the location of the Otemma glacier in Switzerland (red rectangle). Orthoimage provided by swisstopo (2019). Glacier extents adapted from Linsbauer et al. (2016).

## 2.3 Stream discharge

In July 2020, a stream gauging station was installed 100 m downstream of the glacier portal (Fig. 1), in a bedrock-constrained river section to insure the collection of all upstream flow. Stream discharge was estimated using a stage-discharge relationship



(Müller et al., 2022). River stage was measured continuously at 10 minute intervals with a Keller DCX-22AA-CTD datalogger. Discharge was estimated by dilution gauging using Rhodamine WT 20 % dye. The fluorescent dye concentration was measured with a Fluorometer (Albillia GGUN-FL30). Based on 21 gaugings in 2020 and 15 in 2021, the estimated mean discharge uncertainty (95 % confidence) was $\pm 0.55$ m$^3$ s$^{-1}$, but tends to increase for peak discharge with an uncertainty of $\pm 2$ m$^3$ s$^{-1}$

for a river discharge of 13.5 m$^3$ s$^{-1}$. All stream data have already been published (Müller and Miesen, 2022).

## 2.4 Mass balance observations and dye tracing

In-situ monitoring of the seasonal surface mass balance of Otemma glacier was started in 2020 by M. Fischer using the direct glaciological method (Kurzböck and Huss, 2021). For this study, snow depth measurements were performed manually at 5 locations on 26 June 2020 and 92 locations on 28 May 2021 across the entire main lobe of the glacier (from 2560 to 3020 m

a.s.l., Fig. 1). Snow density was estimated on the same dates by measuring the average density of the whole snowpack with a snow sampler in the centre of the main glacier lobe in 2020 and at two locations in 2021 (Fig. 1). Snow water equivalent (SWE) values were calculated by multiplying the measured snow depths with the average density value from the closest snow pit.

End of June 2020, 10 PVC ablation stakes covering the ablation area of the main glacier lobe from an elevation of 2590 to

2890 m a.s.l. (Fig. 1) were installed using a Kovacs ice drill and 8 m drilling rods. Snow and ice melt was measured 3 times in summer 2020 and 7 times in summer 2021. An ice density of 900 g L$^{-1}$ was assumed to convert measured ablation values in ice equivalent to water equivalent.

Between July and August 2020, dye tracing experiments were carried out by injecting Sulforhodamine WT into 6 moulins in the lower part of the glacier, at 370 to 1100 m distance from the glacier portal. The transit time of the dye to the glacier

portal was measured to characterize the englacial water drainage velocity.

## 2.5 Water sampling for stable isotope measurements

Snow for stable isotope analyses was mainly sampled in spring during two periods: from the glacier snout to the highest ablation stake between 24 and 30 June 2020 and across the main lobe of the glacier on 28 May 2021 (Fig. 1). Snow was sampled systematically at various locations by extracting the first five centimeters of the snowpack, which we define as *Snow*

*surface*, and by sampling snow between 10 and 20 cm below the snow surface, called *Snow 10 cm*. Isotopic snow profiles were carried out on 28 May 2021 in two snow density pits, where snow was sampled every 50 cm in depth from the surface to the bottom of the snowpack. The isotopic profile of a third snow pit was sampled just below the glacier terminus on 10 June 2021. There, we also sampled snow melt that formed a thin saturated layer at the base of the snowpack. In summer, the snow surface was sampled on the glacier in mid-July at four locations in 2020 and two locations in 2021. After July, snow only remained on

inaccessible parts of the catchment at high elevation.

Ice samples for isotope measurements were collected at various random locations on the glacier surface during two to four sampling campaigns in each summer from 2019 to 2021. Ice cores were extracted using a manual 20 cm ice screw. On 30 June 2021, two ice cores of 5 and 8 m depth were drilled using a Kovacs ice drill at the location of the 2nd and 8th ice ablation



stakes from the glacier terminus (Fig. 1). Ice was sampled by taking a bulk sample of the ice core every meter. Ice melt water
from small supraglacial channels was also sampled on the glacier, at least one kilometer away from the temporal snowline to
avoid potential mixing with snow melt water. All ice and snow samples were completely melted in a sealed plastic bag in-situ
and then transferred into 12 mL glass vials.

Stream water at the glacier portal was sampled automatically two to three times a day during low and high flows from mid-
June to end of September in 2020 and 2021 using an ISCO 6712 full-size portable water sampler with 24 1L bottles, which
were half filled. Water bottles were transferred to 12 mL glass vials every one to two weeks. The sampler was placed in a
protected, shaded location to avoid water evaporation and the average summer air temperature measured at the nearby weather
station was 7 °C between July and September 2021.

From 2019 to 2021, we collected 39 liquid precipitation samples near the weather station at the glacier terminus (Fig. 1).
Rain water was sampled using a simple PVC funnel, which diverted rain into a plastic bag through a 2 mm plastic tube. All
samples represent the bulk isotopic composition of single rain events and were usually collected the day after the end of a
rain event. We defined rain events as days with rain, separated by at least one day without rain. In winter, we also sampled
fresh snow directly after a few snow events. Due to air temperatures below 0 °C, we assume that little snow transformation or
fractionation occurred, and that these samples therefore represent the isotopic composition of solid precipitation events.

All liquid samples were stored in 12 mL amber glass vials in the field, with air-tight screw caps containing a silicone rubber
septa. Glass vials were flushed with the sample water prior to sample storage to avoid contamination. All water vials were
brought to the laboratory and kept in a cold chamber until analysis. Water stable isotopes were measured using a Wavelength-
Scanned Cavity Ring Down Spectrometer (Picarro 2140i). The median analytical standard deviation of all samples was 0.04
and 0.25 ‰ (maximum standard deviation of 0.11 and 0.65 ‰) for $\delta^{18}$O and $\delta^2$H, respectively. All values are expressed
relative to the international Vienna Standard Mean Ocean Water (VSMOW) standards (Coplen, 1994). All isotopic data as well
as detailed maps of sampling location have been published in Müller (2023a).

## 3 Numerical isotopic and glacio-hydrological modelling

We propose here a framework to model the share of snow melt, ice melt and rain of water reaching the portal of the Otemma
glacier. The model is separated in three main modules, which are calibrated individually (Fig. 2). The first module corresponds
to the mass balance model which simulates snow and ice melt. The mass balance model is validated with the measured stream-
flow at the glacier portal over the observed years. The second module estimates the isotopic composition of each water source
based on the mass balance calculations, and the third module implements a hydrological transfer routine that transfers water
sources simulated for each model cell of the catchment to the glacier portal. All abbreviations, parameters and variables of the
model are summarized in Table A1, including corresponding units.

The model domain, discretized into 200 x 200 m grid cells (total of 586 grid cells), corresponds to the catchment limits
upstream of the glacier portal, where all water drained in the glacierized catchment converges (Fig. 1). For each cell, the mean
elevation, slope and aspect were estimated using a 2 m resolution DEM of 2019 from swisstopo (2019).



**Figure 2.** Schematic representation of the main modelling blocks of the combined isotope and glacio-hydrological model, divided into the independently calibrated main modules. Main calibration parameters are highlighted in red. **(a)** The mass balance module estimates amounts of snow melt, ice melt, rain-on-snow (ROS) and rain for each grid cell. **(b)** The isotopic module uses a calibration curve with air temperature ($T_0$) to estimate $\delta^2$H of precipitation, while $\delta^2$H of ice melt is defined based on ice melt samples on the glacier. **(c)** For each cell, the hydrological transfer module consists of routing the water using a convolution of the water input and gamma distributions ($g(t, \alpha, \beta)$) which represent the travel time distributions of water through four different landcover types in the catchment. Water is finally transferred through a "glacial" fast and slow reservoir. The bottom right figure illustrates the gamma distributions for one specific cell.



## 3.1 Mass balance model

Snow water equivalent (SWE) within the entire catchment was estimated at an hourly time step from October 2019 to October 2021 based on solid precipitation accumulation and snow melt. Snow redistribution, snow sublimation or deposition are also accounted for.

### 3.1.1 Air temperature

For each cell $j$, air temperature ($T_j$) is estimated using measured air temperature ($T_0$) close to the glacier portal (2450 m a.s.l., Fig. 1) and corrected with the mean cell elevation ($z_j$) using a calibrated temperature lapse rate ($\Delta_T$) following Eq. (2).

$$T_j = T_0 - \Delta_T \frac{z_j - 2450}{100} \tag{2}$$

We allow the temperature lapse rate to change seasonally because in alpine glacierized areas higher lapse rates occur in summer compared to winter (Rolland, 2003; Marshall et al., 2007). It is set to resemble a gaussian shaped function depending on the day of the year (DOY) (Eqs. 3 & 4):

$$g(\text{DOY}) = \frac{1}{\sigma_{\Delta_T}\sqrt{2\pi}} e^{-\frac{1}{2}\left(\frac{\text{DOY}-\mu_{\Delta_T}}{\sigma_{\Delta_T}}\right)^2} \tag{3}$$

$$\Delta_T(\text{DOY}) = \frac{g(\text{DOY})}{\max(g(\text{DOY}))} f_{\Delta_T,\text{range}} + f_{\Delta_T,\text{inc}} \tag{4}$$

All four parameters of Eqs. (3) & (4) ($\mu_{\Delta_T}$, $\sigma_{\Delta_T}$, $f_{\Delta_T,\text{range}}$, $f_{\Delta_T,\text{inc}}$) are calibrated for each year. An illustration of the calibrated functions is provided in Fig. D1c.

### 3.1.2 Incoming shortwave radiation

For each model cell, the corrected incoming shortwave radiation ($I_{\text{corr}}$) is estimated based on the measured incoming shortwave radiation measured at the weather station ($I_0$) by taking into account the terrain slope ($\theta$ in degree) and aspect ($\gamma$ in degree) (Eq. 5). First, the radiation is increased with steeper slope by a certain factor ($f_{\text{rad,slope}}$) until a maximum slope threshold ($\theta_{\text{max,rad}}$) is reached (Eq. 6). Then, a factor ($\gamma_{\text{max,rad}}$) is subtracted to account for aspect. It corresponds to 0 when aspect is 180 $^\circ$ (south-facing slopes) and is increased linearly with terrain aspect facing north until it reaches a maximal calibrated factor ($\gamma_{\text{max,rad}}$). This factor is then scaled with slope, so that steep cells are more affected by aspect than flatter cells (Eq. 7). An illustration of the resulting function is provided in Fig. D1d.





$$I_{\text{corr}} = I_0(f_{\text{rad,slope,tot}} - f_{\text{rad,aspect,tot}}) \tag{5}$$

$$f_{\text{rad,slope,tot}} = 1 + \cos\left(\frac{90}{\theta_{\text{max,rad}}}(\theta - \theta_{\text{max,rad}})\right)f_{\text{rad,slope}} \tag{6}$$

$$f_{\text{rad,aspect,tot}} = \gamma_{\text{max,rad}}\frac{|\gamma - 180|}{180}\sin(\theta) \tag{7}$$

### 3.1.3  Liquid and solid precipitation estimation

The temperature thresholds to separate liquid from solid precipitation are set to a lower value of 1 °C (below only snow) and an upper value of 2 °C (above only rain), with a linear fraction in between. For liquid precipitation, we use rain measurements ($P_0$) from the weather station at the glacier portal (2450 m a.s.l., Fig. 1). For solid precipitation, snowfall is inferred from the nearest automatic weather stations ($P_{\text{meteo}}$) located in Otemma for 2019/2020 and Arolla for 2020/2021 (cf. Sect. 2.2). A fixed snow correction factor ($f_{\text{corr,snow}}$) for the whole winter is calibrated for each year (Eq. 8). We then correct precipitation ($P_j$) for each cell $j$ with elevation ($z_j$) based on a precipitation lapse rate ($\Delta_P$) calibrated for each year.

$$P_j = \begin{cases} P_0\left(1 + \Delta_P\frac{z_j - 2450}{100}\right) & \text{if } T_0 > 1°\text{C} \\ f_{\text{corr,snow}}P_{\text{meteo}}\left(1 + \Delta_P\frac{z_j - 2450}{100}\right) & \text{if } T_0 \leq 1°\text{C} \end{cases} \tag{8}$$

### 3.1.4  Snow redistribution

We account for snow redistribution based on terrain slope ($\theta$) by defining a calibrated slope threshold ($\theta_{\text{redist,thresh}}$) above which snow redistribution occurs (Eq. 9). Above this threshold, we decrease the amount of solid precipitation ($P_j$) received by each model cell by a certain factor ($f_\theta$). We then redistribute the corresponding total amount to all other cells by defining a redistribution function which uses an increase factor ($f_{\text{redist}}$) for solid precipitation (Eq. 9).

$$P_{\text{sf}} = \begin{cases} P_j(1 - f_\theta(\tan(\theta - \theta_{\text{redist,thresh}}))) & \text{if } \theta > \theta_{\text{redist,thresh}} \\ P_j f_{\text{redist}} & \text{if } \theta \leq \theta_{\text{redist,thresh}} \end{cases} \tag{9}$$

The value of $f_{\text{redist}}$ for each cell is calibrated by defining a calibration objective function where the total monthly amount of solid precipitation removed from steep slopes ($\theta > \theta_{\text{redist,thresh}}$) equals the monthly total amount redistributed to all other cells based on their elevation. This method conserves the total mass of solid precipitation in a simple way without an estimation of curvature of connected cells, and compensates for local anomalies between observed and modelled SWE. An illustration of the resulting functions is provided in Fig. D1a & b.

### 3.1.5  Snow and ice melt

Snow melt is estimated using an enhanced temperature-index melt model (Gabbi et al., 2014) (Eq. 10). Ice melt at the glacier surface was estimated using the same equation with different parameter values.



$$M = \begin{cases} f_{\mathrm{melt,T}}T_j + f_{\mathrm{melt,rad}}(1 - \alpha_{\mathrm{snow}})I_{\mathrm{corr}} & \text{if } T_j > T_{\mathrm{melt}} \\ 0 & \text{if } T_j \leq T_{\mathrm{melt}}, \end{cases} \tag{10}$$

$$\alpha_{\mathrm{snow}} = 0.86 - 0.155\, log_{10}(T_{acc}) \tag{11}$$

where $I_{\mathrm{corr}}$ is the corrected incoming shortwave radiation (see also Eq. 5) and $T_j$ the air temperature for the cell $j$. Snow albedo ($\alpha_{\mathrm{snow}}$) was estimated following the work of Gabbi et al. (2014). It is assumed to decrease from a value of 0.86 as a function of the accumulated daily maximum positive air temperature ($T_{acc}$) since the last snowfall (Eq. 11). The threshold
temperature ($T_{\mathrm{melt}}$) distinguishing between melt and no melt is a calibration parameter. The temperature melt factor ($f_{\mathrm{melt,T}}$) as well as the shortwave radiation factor ($f_{\mathrm{melt,rad}}$) were calibrated for snow and ice separately. The albedo of ice is set to 0.25 (Gabbi et al., 2014).

### 3.1.6 Sublimation and deposition

An estimation of the snow sublimation rate was required in this work since it may significantly impact the snowpack isotopic
signature due to fractionation (Ala-Aho et al., 2017). Sublimation is estimated following Todd Walter et al. (2005) based on the difference of vapor density between the snow surface and the air divided by the resistance to vapor exchange, which requires wind speed data and, in particular, an estimation of the snow surface temperature ($T_{\mathrm{sp}}$). Wind speed is roughly estimated using data from the most nearby automatic weather station (see Sect, 2.2). Since no snow surface temperature data are available, we propose to estimate $T_{\mathrm{sp}}$ by first defining snow surface temperature similar to air temperature ($T_j$). Then, we estimate a
simplified snow surface energy balance ($E_{\mathrm{net}}$) based on its two main terms: net shortwave ($S_{\mathrm{net}}$) and net longwave ($L_{\mathrm{net}}$) radiation (Stigter et al., 2021). When $E_{\mathrm{net}}$ is positive, usually due to the atmospheric shortwave radiation during clear-sky days, we assume that air and snow temperature are similar. When $E_{\mathrm{net}}$ is negative, usually during clear-sky nights when outgoing longwave radiation from the snow surface is the major energy flux, $T_{\mathrm{sp}}$ cools more than air. This cooling effect is calibrated by a temperature factor ($f_E$) following Eqs. (12) to (16). During cloudy nights and days, $E_{\mathrm{net}}$ usually remains close to zero due to
limited incoming shortwave radiation (but increased atmospheric longwave radiation) and $T_{\mathrm{sp}}$ is close to $T_j$. More details on the snow energy balance and snow surface temperature can be found in the work of Stigter et al. (2021).

$$T_{\mathrm{sp}} = \min(T_j + E_{\mathrm{net}}f_E \; ; 0^\circ C) \tag{12}$$

$$E_{\mathrm{net}} = \min(S_{\mathrm{net}} + L_{\mathrm{net}} \; ; 0 \, \mathrm{J\, m^{-2}}) \tag{13}$$

$$S_{\mathrm{net}} = (1 - \alpha_{\mathrm{snow}})I_{\mathrm{corr}} \tag{14}$$

$$L_{\mathrm{net}} = \epsilon_{\mathrm{air}}\sigma T_j^4 - \epsilon_{\mathrm{snow}}\sigma T_j^4 \tag{15}$$

$$\epsilon_{\mathrm{air}} = (0.72 + 0.005T_j)(1 - 0.84f_{\mathrm{cloud}}) + 0.84f_{\mathrm{cloud}} \tag{16}$$





where $\sigma$ is the Stefan–Boltzmann constant and $\epsilon_{\mathrm{snow}} = 0.97$ is the emissivity of the snow surface. The emissivity of air ($\epsilon_{\mathrm{air}}$) is estimated based on the fraction of cloud cover ($f_{\mathrm{cloud}}$). The fraction of cloud cover was assessed by dividing the measured shortwave radiation ($I_0$) with the theoretical maximal shortwave radiation. $f_{\mathrm{cloud}}$ was set to 1 when less than 50 % of theoretical 265 maximal shortwave radiation was measured at the weather station and was set to 0 otherwise (Todd Walter et al., 2005).

### 3.1.7 Calibration

Prior to calibration of the mass balance model, we initialised the model values for SWE and $\delta^2$H by first running the model for one year with initial SWE = 0 for all cells (and uncalibrated model parameter values).

Mass balance model parameters calibration was then performed using PEST-HP. This model-independent algorithm itera-270 tively minimizes the variance of the error between model outputs and corresponding field observations via inverse estimation (Doherty, 2015). We defined three data sets of field observations. The first corresponds to the measured end-of-winter SWE on the main lobe of the glacier. The second data set corresponds to the annual ice melt measured at the ablation stakes at the end of each summer (Fig. 1). The third data set of field observations corresponds to maps of the temporal snow cover during the entire ablation periods. We used daily 3 m resolution Planet satellite imagery (PlanetScope Scene (Team Planet, 2017)) and 275 manually identified clear sky days during the summer months of 2020 and 2021. We then automatically identified snow cover using a K-Means unsupervised learning algorithm from Google Earth Engine (Arthur and Vassilvitskii, 2007) and created, for approximately every second week during the melting seasons, maps of snow presence/absence for our discretised model domain. We set PEST-HP to minimize the error between modelled and observed snow presence/absence for each grid cell and all available snow maps. This procedure leads to a better determination of the temporal evolution of the snowline and should 280 improve the modelled SWE estimation especially in zones where no direct SWE observations are possible (Barandun et al., 2018).

All 26 calibration parameters were calibrated separately for each hydrological year (starting October 1st) but the calibration procedure was performed by simulating both years at once and calibrating twice all parameters, so that SWE and snow cover simulated at the end of the first year are used as initial values for the second simulation year. Table A1 summarizes the results 285 of the calibration procedure.

### 3.2 Snow isotopic module

### 3.2.1 Basic model formulation

Due to the strong correlation between water stable isotopes of oxygen ($\delta^{18}$O) and deuterium ($\delta^2$H), we base the rest of this study on $\delta^2$H only.

Using SWE values from the mass balance model, we estimate the mean snowpack isotopic composition ($i_{\mathrm{sp}}$) for each model cell (Fig.2). The same approach as proposed by Ala-Aho et al. (2017) is used to estimate the isotopic composition of the snowpack and of snow melt over time. An amount-weighted approach based on a precipitation input in the form of rain ($P_{\mathrm{r}}$) or snowfall ($P_{\mathrm{sf}}$), snow sublimation ($E_{\mathrm{sp}}$) and snow melt ($M_{\mathrm{snow}}$) is applied (Eq. 21). A simple fractionation routine is used





for snow melt ($i_{sm}$) and snow sublimation ($i_E$) using two calibration parameters ($f_{frac,sm}$, $f_{frac,E}$) and $n_{melt}$, the number of days
295 since the beginning of snow melt (Eqs. 17 & 18).

$$i_{sm} = i_{sp} - \frac{f_{frac,sm}}{n_{melt}} \tag{17}$$

$$i_E = i_{sp} - f_{frac,E} \tag{18}$$

$$i_r = a_r + b_r T_0 \tag{19}$$

$$i_{ROS} = i_{sp} f_{ROS} + i_r (1 - f_{ROS}) \tag{20}$$

$$i_{sp}^t = \frac{i_{sp}^{t-1} h_{SWE}^{t-1} + i_r^t f_{ROS} P_r^t + i_{sf}^t P_{sf}^t - i_E^t E_{sp}^t - i_{sm}^t M_{snow}^t}{h_{SWE}^{t-1} + f_{ROS} P_r^t + P_{sf}^t - E_{sp}^t - M_{snow}^t} \tag{21}$$

The year-round isotopic composition of the precipitation as rain ($i_r$) or snowfall ($i_{sf}$) is determined by computing a linear
regression curve between the measured air temperature at the weather station ($T_0$) and the isotopic composition of sampled
precipitation events (Eq. 19, see also Sect. 3.2.2).

### 3.2.2 Air temperature and precipitation stable isotopes relationship

305 To estimate the snowpack isotopic composition, we relate the isotopic composition of each precipitation event to air temper-
ature. For each precipitation sample, the corresponding temperature of the event is estimated by calculating the average air
temperature weighed by the amount of precipitation measured each 10 minutes during the previous day. No clear trend in $\delta^2 H$
with elevation could be observed from rain samples obtained 8 times during the melt season at both 2450 and 2800 m a.s.l.
For this reason, no isotopic lapse rate is used and the isotopic composition of precipitation events was similar for the entire
310 catchment. This choice is discussed in Sect. 5.2.

In order to assess the uncertainty in the relationship between air temperature and precipitation $\delta^2 H$, we define a normally
distributed error for both parameters. We apply a Gaussian distribution with a standard deviation ($\sigma$) of 1 °C for air temperature
and 5 ‰ for $\delta^2 H$. We then perform 5000 iterations for which we randomly picked values in their distributions and then
calculated a linear fit each time. These 5000 realizations are assumed to represent the uncertainty range of this relationship.
315 This uncertainty margin is used to provide a sensitivity analysis of the impact of the air temperature and precipitation $\delta^2 H$
relationship on the isotopic snow melt model results.

### 3.2.3 Rain-on-Snow (ROS)

ROS incorporation in the snowpack and its water release is a complex process, which may have a strong impact on the snowpack
isotopic composition, depending on whether rain water leaks through the snowpack, is stored or refreezes in the snowpack
320 (Juras et al., 2017; Beria et al., 2018). The isotopic model from Ala-Aho et al. (2017) assumes a complete incorporation of the
rain in the snowpack, however field-based studies have highlighted partial mixing of rain and snowmelt in the snowpack (Juras
et al., 2017; Rücker et al., 2019a). To account for the latter, we introduce a factor ($f_{ROS}$) which defines the fraction of rain water



which is incorporated in the snowpack and which contributes to modifying its isotopic composition (Eq. 20). As observed in the work of Juras et al. (2017), we assume that the total amount of ROS event water is released from the snowpack, but part

of this release (equivalent to $f_{\text{ROS}}P_{\text{ROS}}$) is composed of previously stored snow melt water pushed out of the snowpack. The isotopic composition of the rain water leaving the snowpack ($i_{\text{ROS}}$) is then composed of a mix of snow melt water and rain water (Eq. 20). We defined a simple calibration function between SWE and $f_{\text{ROS}}$, where $f_{\text{ROS}}$ increases with lower SWE. This relationship is based on the assumption that, during the melting season, thicker snowpacks are less ripe (due to less melt) and water infiltration is faster because of more preferential flow paths, so that the incorporation of rain water in the snowpack is

only partial. In thinner snowpacks, we assume that snow is ripe, which leads to more rain water trapped and mixed within the snowpack.

### 3.2.4 Snow isotopic module calibration

The three isotope parameters ($f_{\text{frac,E}}$, $f_{\text{frac,sm}}$, $f_{\text{ROS}}$) were calibrated manually following simple rules. First, the resulting snow melt $\delta^2$H value should remain below the measured stream $\delta^2$H during the early melting season since snow melt is isotopically

lighter than ice melt (snow melt is the end-member with the most negative values). Secondly, at the onset of snow melt when snow still covers the glacier (in June in this work), snow melt $\delta^2$H should be close to stream $\delta^2$H since snow melt is the major contributor at that time. Then, the modelled snowpack $\delta^2$H at a grid cell should also be similar to the measured depth-averaged isotopic composition of the corresponding snowpit (Fig. 1). Finally, during the first half of the melting season (July here), we observed that daily isotopic variations in the stream showed a minimum (more depleted) in the morning and a maximum (more

enriched) during the afternoon. This is likely due to an increased contribution from ice melt in the afternoon compared to snow melt since ice melt shows a less depleted $\delta^2$H value than snow melt at that time. During the late melting season (mid/end August to September), the stream isotopic minimum occurs in the afternoon, indicating here that the snow melt isotopic composition has become more enriched than ice melt, due to snow fractionation and rain-on-snow. The moment when the stream $\delta^2$H signal switches from a minimum in the morning to a minimum in the afternoon indicates when snow melt $\delta^2$H becomes isotopically

heavier than ice melt.

### 3.3 Hydrological transfer module

A simple hydrological transfer scheme is used to transfer water with its isotopic composition from its input grid cell to the catchment outlet. In this module, we do not consider any interactions between hydrologically connected cells but only use the hydrological path length from each cell to the catchment outlet. We divided the hydrological paths in four different categories:

(1) flow through the snowpack; (2) flow through the hillslope sediments (if outside of the glacier); (3) flow through the en-/subglacial distributed system, and (4) flow through the en-/subglacial channelized system. The total flow path from each grid cell to the glacier portal was calculated using the Flow Distance tool (ArcGIS Pro v2.3) and a 2 m DEM of 2019 (swisstopo, 2019). For each category, we apply a convolution between the water input at time $t$ and a time-dependent gamma distribution probability density function ($g(t, \alpha_g, \beta_g)$) as described in Eqs. 22 & 23. Here, the gamma function is used to reproduce a





realistic transit time distribution (TTD) of the water input (McGuire and McDonnell, 2006). The convolution of the TTD at
each time step provides the total TTD of the water from each grid cell to the glacier portal (Fig. 2).

$$\delta_{out}(t) = \int_0^\infty g(\tau)\delta_{in}(t-\tau)d_\tau = g(t) * \delta_{in}(t),\tag{22}$$

$$g(t,\alpha_g,\beta_g) = \frac{\beta_g^{\alpha_g} t^{(\alpha_g-1)} e^{-\beta_g t}}{\Gamma(\alpha_g)},\tag{23}$$

where $(\delta_{in}(t))$ is any given input of water at time $t$ and $(\delta_{out}(t))$ is the output water flux.

To estimate the TTD, the parameters of the gamma distribution need to be defined. For each of the four hydrological flow
categories, we estimate the mean transit time ($t_{MTT}$) of the water based on physical properties of each category and use this
travel time to define the mode of the gamma distribution ($t_{MTT} = \frac{\alpha_g-1}{\beta_g}$). The dispersion of the flow for the gamma distribution
is defined by a dispersion factor ($D = \alpha_g$-1).

### 3.3.1 Hillslope routing

We assume that the landcover outside glacierized areas is mainly composed of hillslope sediments. In some steep parts, bedrock
is apparent and therefore the estimated transit time may be somewhat slower than in reality as water flows faster on the bedrock.
However, this simplification should not lead to a large bias as sediments still dominate the hillslopes (Müller et al., 2022).
For the latter, the average transit time ($t_{MTT}$) from each grid cell to the glacier surface was calculated using an estimated
groundwater pore velocity (Eq. 24). Pore velocity is defined for kinematic subsurface saturated flow (MacDonald et al., 2012)
as a function of slope ($\theta$), aquifer distance ($L_a$), aquifer porosity ($\eta$) and hydraulic conductivity ($K_s$). We selected a porosity
of 0.3 and a hydraulic conductivity for talus slopes of $5\times10^{-2}$ m s$^{-1}$ based on previous research (Müller et al., 2022).

$$t_{MTT} = \frac{L_a}{v_p} = \frac{L_a\eta}{K_s\sin(\theta)}\tag{24}$$

### 3.3.2 Snowpack routing

For snow melt or ROS events, a TTD through the snowpack is used. Here, we calibrated an average pore velocity in the
snowpack with an initial velocity of 1200 mm h$^{-1}$ following Juras et al. (2017). As for hillslope, the average transit time
($t_{MTT}$) through the snowpack is used to define the mode of the gamma distribution. Since SWE evolves with time, the TTD
through the snowpack changes with time and was recalculated for each day.

### 3.3.3 Glacier routing

Once the flowpath reaches the glacier ice surface, we defined two different water flow categories. The en-/subglacial drainage
system was considered to be either distributed or channelized. During the winter, less melt and creep closure occurs due to ice
dynamics (Flowers, 2015). Subglacial channels tend to close, leading to an inefficient distributed drainage system characterized





by slow water flow. During summer, larger conduit-like subglacial channels tend to develop and extend up-glacier with the recession of the temporal snowline (Nienow et al., 1998). Therefore, based on the temporal snow cover estimated with the mass balance model, we calculated the mean distance between the glacier portal and the first 5 grid cells on the glacier with
snow cover to define the length of the channelized flow. We neglect supraglacial meltwater runoff here. However, the calibration of the glacier routing may compensate for this simplification by artificially increasing the subglacial velocity. The length of the channelized flow and the corresponding TTD for each grid cell changes through time since it is based on the temporal snow cover evolution. The length of the distributed flow is computed as the difference between the total flow length on the glacier and the channelized flow length.

For the snow-covered distributed en-/subglacial system, the mean velocity could not be measured directly. Here, an initial value of 0.1 m s$^{-1}$ was used following Nienow et al. (1998). For the summer channelized system, the velocity was defined based on 25 dye tracing injections. Measured velocity ranged between 0.29 and 0.83 m s$^{-1}$ and a velocity of 0.8 m s$^{-1}$ was selected to represent a fully channelized system.

### 3.3.4  Total runoff transfer to the outlet

The convolution of the combined gamma distributions (snow, hillslope, distributed subglacial drainage, channelized subglacial drainage) with the different water sources time series (rainfall ($P_{r,j}$), ROS, ($P_{ROS,j}$), snow melt ($M_{snow,j}$), or ice melt ($M_{ice,j}$)) obtained from the mass balance model for each grid cell $j$ with an area ($A_j$) provides the estimated discharge contribution at the catchment outlet per water source and per cell. The sum over all grid cells corresponds to the total discharge from each water source. The case for snow melt is illustrated in Eqs. 25 & 26.

$$Q_{sm,tot} = \sum_{cell=1}^{cell=n} \Big( g_j(t, \alpha_g, \beta_g) * (M_{snow,j} A_j) \Big) \tag{25}$$

where $Q_{sm,tot}$ is the total discharge from snow melt at the catchment outlet. The same approach can be applied to estimate the mean isotopic composition of the other water sources by applying the same convolution to the multiplication of the precipitation or melt time series and the isotopic signal ($i_{sm}$, $i_{ROS}$ and $i_r$). This assumes that the isotopic composition of a water input is transported and redistributed to the catchment outlet following the same TTDs. The sum of each grid cell divided by the total
discharge corresponds to the amount-weighted average isotopic composition of the corresponding water source (Eq. 26 for the case of snow melt).

$$i_{sm,tot} = \frac{\sum_{j=1}^{j=n} \Big( g_j(t, \alpha_g, \beta_g) * (i_{sm,j} M_{snow,j} A_j) \Big)}{Q_{sm,tot}} \tag{26}$$

### 3.3.5  Fast and slow glacier storage

It is likely that part of the water is temporally stored in some areas of the en-/subglacial drainage network. To account for this,
we ultimately define two reservoirs which represent a fast and slow linear storage. The integrated discharge of all water sources

//doi.org/10.5194/egusphere-2024-631




after the convolution with the gamma distributions is then separated between both reservoirs based on a calibrated fraction ($f_{\text{reservoir}}$), which assigns how much goes into the slow reservoir. The outflow discharge of each reservoir finally depends on a calibrated response time constant ($k$). For the fast reservoir, this results in (Eq. 27).

$$Q_{\text{fast}} = \frac{S_{\text{fast}}}{k_{\text{fast}}}, \tag{27}$$

where $S_{\text{fast}}$ is the filling of the reservoir. The slow reservoir response is computed in analogy to the equation above. The isotopic composition of each reservoir is separated between all water sources and is assumed to be fully mixed at each time step.

### 3.3.6 Hydrological transfer calibration

The calibration of the entire hydrological transfer module (including hillslope routing, snow routing, routing of distributed subglacial drainage and of channelized subglacial drainage and transfer via two linear reservoirs) was also performed using the
optimisation algorithm PEST-HP, as already proposed in other glacio-hydrological studies (e.g., Immerzeel et al., 2012). We set three objective functions. The first function minimizes the error between observed and simulated discharge at the catchment outlet at an hourly time step. The second function optimizes the amplitude of daily stream discharge variations, which is a typical feature of glacier streams and which tends to increase during the summer season (Nienow et al., 1998; Lane and Nienow, 2019). Finally, the last objective function aims to minimize the difference between the observed and modelled $\delta^2$H of
the stream.

We finally tested the differences in model performance by using only the discharge data or by adding the water isotopes for the calibration. The calibration was performed by only including data for the summer 2020 (26 June to 15 September) and 2021 (8 June to 20 June). The first two weeks of June 2021 were included as they were not recorded in 2020 and represent the initial significant increase in streamflow after winter (when discharge becomes larger than $1 \text{ m}^3 \text{ s}^{-1}$). The period from 20 June
2021 to 15 September 2021 was then used to evaluate the model performance.

### 3.4 Mixing model for water sources using water stable isotopes

In order to estimate the contribution from rain, snow and ice melt, a three components mixing model needs to rely on two independent tracers. Here, since we only rely on water stable isotopes, we propose estimating the shares of rain ($\phi_{\text{rain}}$) and rain-on-snow ($\phi_{\text{ROS}}$) using the output discharge of the hydrological transfer module divided by the total modelled discharge at
the glacier portal. Then, we only use isotopes to estimate the share of snow ($\phi_{\text{snow}}$) and of ice melt ($\phi_{\text{ice}}$) following Eq. 28. The isotopic composition of rain ($i_{\text{r}}$) and snow ($i_{\text{sm}}$) is estimated using the isotopic model. The isotopic composition of ice melt ($i_{\text{ice}}$) is defined as constant through the year based on our measurements.

$$\phi_{\text{ice}} = \frac{i_{\text{stream}} - i_{\text{sm}} - (i_{\text{r}} - i_{\text{sm}})\phi_{\text{rain}} - (i_{\text{ROS}} - i_{\text{sm}})\phi_{\text{ROS}}}{i_{\text{ice}} - i_{\text{sm}}} \tag{28}$$





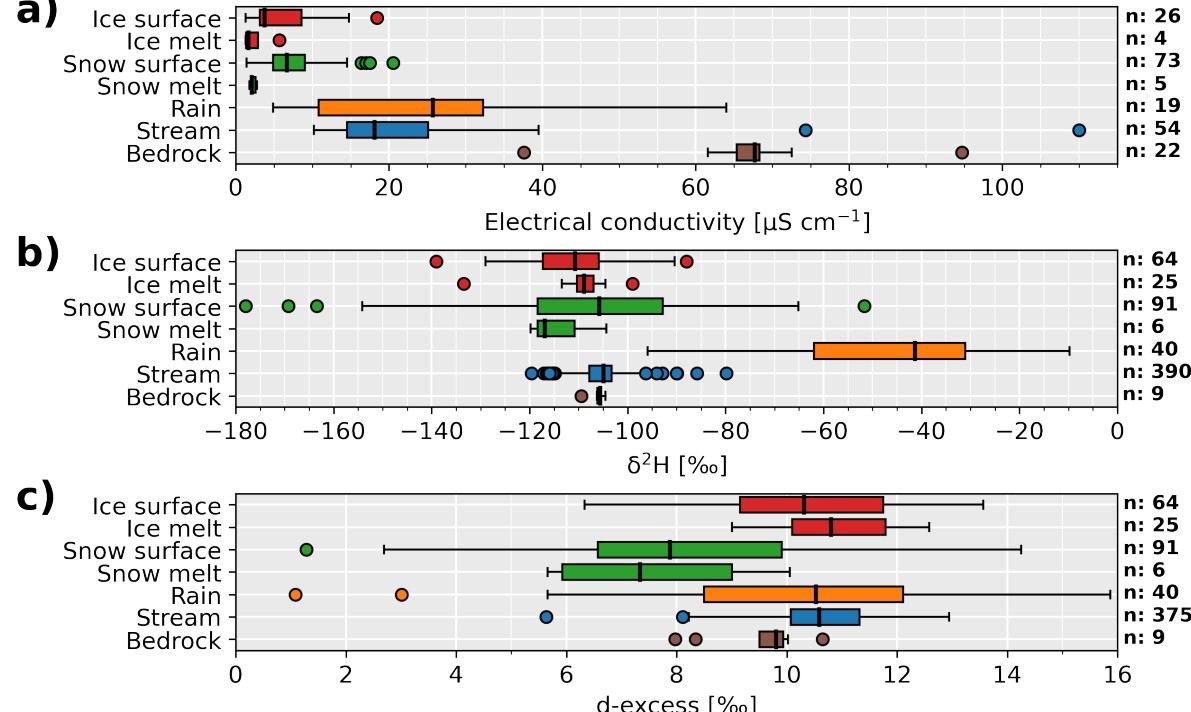

**Figure 3.** Boxplots of all water sources collected between July 2019 and October 2021 including water exfiltrations from the bedrock sidewalls. **(a)** Measured water electrical conductivity; **(b)** $\delta^2$H of water stable isotopes and **(c)** corresponding d-excess. The total number of samples (n) is indicated on the right.

## 4 Results

### 4.1 Isotopic measurements in the field


Between July 2019 and October 2021, we measured the water $\delta^2$H as well as the water electrical conductivity (EC) at different locations within the catchment. We summarize all results in Fig. 3. The rain water EC has a median value of 26 $\mu$S cm$^{-1}$. Snow and ice samples have lower EC values, usually below 10 $\mu$S cm$^{-1}$, likely due to the preferential elution of solutes in the snowpack (Costa et al., 2020). Interestingly, the glacial meltwater stream shows systematically higher EC values than the snow

and ice samples, even during the peak snow and ice melt period. Regarding water stable isotopes, only rain has a significantly different composition (Fig. 3). The surface snow and ice samples have similar $\delta^2$H ranges and show a large scatter, which completely overlap with the stream signal. As suggested in other studies (e.g., Beria et al., 2018), the composition of the snow melt samples shows less scatter than the snow surface samples. In addition to $\delta^2$H , we also report d-excess (d-excess = $\delta^2$H - $8\delta^{18}$O ). The isotopic signature of global precipitation follows a linear relationship called the global meteoric water line with

$\delta^2$H = $8\delta^{18}$O + 10. During non-equilibrium processes such as snow sublimation, the snowpack becomes more rapidly enriched





in $^2$H than $^{18}$O, leading to a decrease in the d-excess value compared to its reference value of 10 (see Beria et al. (2018) for a more complete review). Therefore, the snow surface and snow melt d-excess values appear lower than the other water sources (Fig. 3).

Close to the date of maximum end-of-winter snow accumulation on 28 May 2021, snow surface samples show a large $\delta^2$H

variability with no clear tendency with elevation (Fig. 4a). Snow samples at the same locations but at a depth of 10 cm have different values with no clear trend. The snowpack $\delta^2$H profiles appear stratified with a tendency towards more isotopically depleted snow with depth, reflecting the colder air temperature of the snowfall in the early winter season, which was conserved in the snowpack. D-excess at the snow surface shows no particular trend with elevation but appears lower than for samples at 10 cm depth (Fig. 4c). This is likely due to snow sublimation at the surface. The d-excess of the snow profiles shows a more

significant trend with elevation, likely due to more evaporative fractionation at low elevation and potential condensation and deposition of depleted water vapor at higher elevation. A d-excess lapse rate with elevation has been reported for other high mountain catchments (Carroll et al., 2022b; Sprenger et al., 2023).

From all snow samples collected each year, no significant seasonal trend can be observed (Fig. C1). It was expected that the snowpack would become gradually more enriched in heavier isotopes due to ROS events (Juras et al., 2017) and fractionation

from snow sublimation and melt (Beria et al., 2018). Such behavior is difficult to observe for surface snow samples since the spatial variance of their isotopic composition is large and not representative of the whole snowpack composition.

The $\delta^2$H of the surface ice shows a smaller variability than surface snow and no trend with elevation (Fig. 5). Superficial ice melt samples show the least scatter likely due to more mixing of the ice melt water at the surface (Fig. 5). Two ice cores reaching 5 and 8 meters below the ice surface were also analyzed and show limited variations in $\delta^2$H with depth, while their

average value is close to the ice melt samples. There seems to be a more significant trend in d-excess for the ice cores with elevation but this trend relies only on two sampling locations.

### 4.2   Mass balance model calibration results

The mass balance model parameters were calibrated for both years against measured SWE, measured ice melt and mapped seasonal snow cover (Table A1). The calibrated temperature lapse rate shows a maximum around late May to early June,

a mean value of 0.42 and 0.48 °C per 100 m for 2020 and 2021, and a maximal seasonal variation of 0.18 °C per 100 m (Fig. D1c). Regarding snow redistribution, both calibration years show a similar slope correction factor, with a slope threshold of 32 ° above which snow redistribution occurs towards gentler slopes (Fig. D1a). The calibration of the snow redistribution is relatively similar for both years, with some redistribution near the glacier terminus and above an elevation of 2800 m a.s.l. (Fig. D1b). An important snow redistribution occurs only in 2021 at high elevations (3400 to 3600 m a.s.l.), which mainly

corresponds to few very small high elevated hanging glaciers, where snow redistribution from the nearby steep rock walls is likely. These elevated zones represent a limited total area of the catchment (3 %) so that the impact on the total mass balance is limited. The radiation correction factor varies between 1 for flat slopes and 2.5 for south-facing slopes around 60° (Fig. D1d). Finally, a precipitation lapse rate of 2.2 and 2.6 % per 100 m for 2020 and 2021 was found (Table A1).





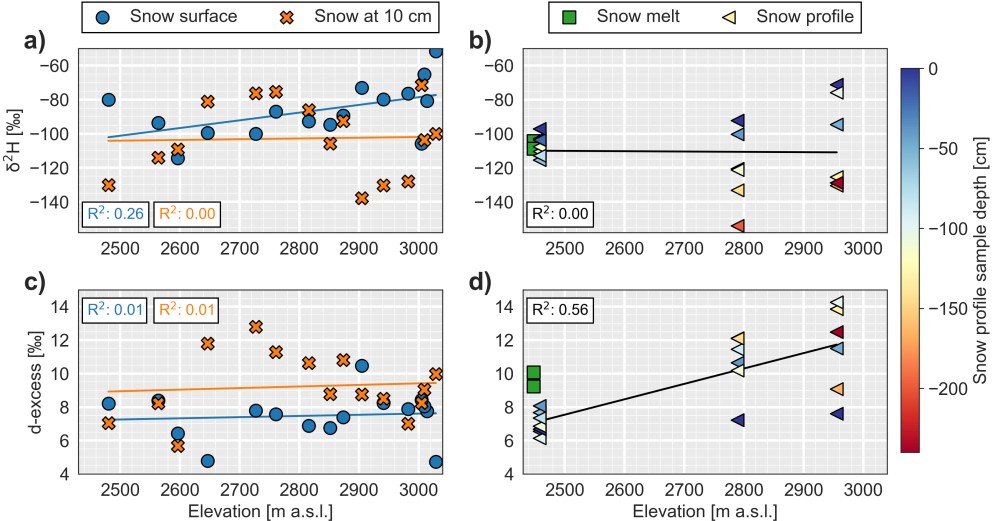

**Figure 4.** Isotopic composition of snow samples with elevation collected on 28 May 2021 on the main glacier lobe (Fig. 1). **(a)** Snow samples collected at the surface and at the same location between 10 and 20 cm depth. **(b)** Snow profiles with sampling depth (0 to -250 cm) indicated by the colorbar. The snow melt found at the bottom of the lowermost profile is also indicated (green rectangles). **(c)** & **(d)** Corresponding results for d-excess. Linear regression curves for sample class with their coefficient of determination ($R^2$) are also shown.

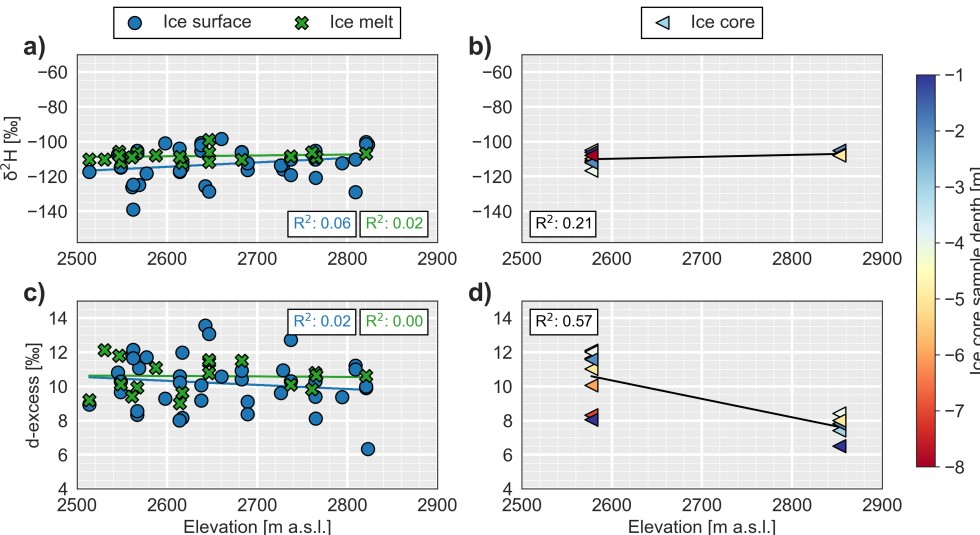

**Figure 5.** Isotopic composition of ice samples with elevation collected between 2020 and 2021 on the main glacier lobe (Fig. 1). The scale of the y-axes is similar to Fig. 4 for comparison. **(a)** Solid ice samples collected at the glacier surface and ice melt water samples collected in gullies at the glacier surface. **(b)** Ice core samples with sampling depth (-1 to -8 m) indicated by the colorbar. **(c)** & **(d)** Corresponding results for d-excess. Linear regression curves for each sample class with their coefficient of determination ($R^2$) are also shown.



Overall, the model shows good performance for SWE, although the model results show less variation with elevation than the
point SWE measurements which are more spatially variable (Fig. E1). The root mean square errors (RMSE) for SWE are 97.9
and 100.3 mm over the hydrological years (starting October 1st) 2019/2020 and 2020/2021 respectively. The RMSE values
for ice ablation are 237.5 mm (in water equivalent, hereafter w.e.) for 2019/2020 and 263.8 mm w.e. for 2020/2021. The mean
error of the snow and ice mass balance calculations is close to 0 mm w.e, except for the snow mass balance in 2020, for which
the model seems to overestimate SWE with a mean error of 45.7 mm. The mapped temporal snow cover evolution is well
represented by the model during the entire melting season, showing similar patterns of melt, with earlier snow disappearance
on steep south-facing slopes (Fig. E2 to E5). In 2020, one summer snow event seems underestimated by the model, leading to
a constant bias in the modelled snow cover fraction after July 2020 (Fig. E6). In 2021, the modelled snow cover evolution fits
well with the mapped extents during the whole melting season, with a somewhat earlier snow disappearance at high elevation,
potentially due to precipitation underestimation in this zone.

Catchment-wide average snow melt over the hydrological years is 1860 and 1527 mm w.e. for 2019/2020 and 2020/2021,
and 1265 and 958 mm w.e. for ice melt. Catchment-wide liquid precipitation amounts to 227 and 320 mm, snow sublimation
to 84 and 82 mm w.e., and snow deposition to 34 and 14 mm w.e. for 2019/2020 and 2020/2021. Although sublimation losses
were simulated with a simple routine, modelled amounts and rates are in a similar range compared to other studies in high
mountain catchments (e.g., Strasser et al., 2008; Stigter et al., 2018).

Finally, the results of the modelled mass balance losses (through rainfall, snow melt and ice melt) at a daily timescale
appear to match well with the measured discharge at the glacier portal (Fig. D2). In particular, the cumulative mass balance
follows well the cumulative measured discharge, except for September 2020, when the modelled mass balance is overestimated
compared to the measured discharge.

### 4.3 Hydrological transfer module results

The calibration of the hydrological transfer parameters was performed in a second separate step following the mass balance
model calibration. Parameters are summarized in Table A1.

Hillslope parameters calibration leads to a lower $K_s$ value than initialized, which significantly smoothes out the response
from the hillslope water inputs. The channelized en-/subglacial routing shows a similar calibrated velocity ($v_{\text{channelized}}$) as
measured, with limited dispersion ($D_{\text{channelized}}$, dispersion decreases with increasing value). The distributed en-/subglacial
routing has a slower velocity ($v_{\text{distributed}}$) than the channelized system, with more dispersion ($D_{\text{distributed}}$). The rain and snow
melt infiltration ($v_{\text{ROS}}, v_{\text{sp}}$) through the snowpack is fast and leads to a transit time of less than an hour for all model grid cells.
The parameter estimation of the fast reservoir indicates that it collects 55 % of all water and has a response time constant ($k_{\text{fast}}$)
of 1 hour. The rest of the inflow is stored in a slow reservoir for which a response time ($k_{\text{slow}}$) of 51 hours is estimated via
calibration.

The final discharge results are shown in Fig. 6. The routing time estimated from $D_{\text{channelized}}$ and $D_{\text{distributed}}$ combined with
the seasonal evolution of their water flowpath length results in an accurate representation of the increase in daily discharge
fluctuations occurring during the melting season as discussed by Lane and Nienow (2019). Discharge recessions during short





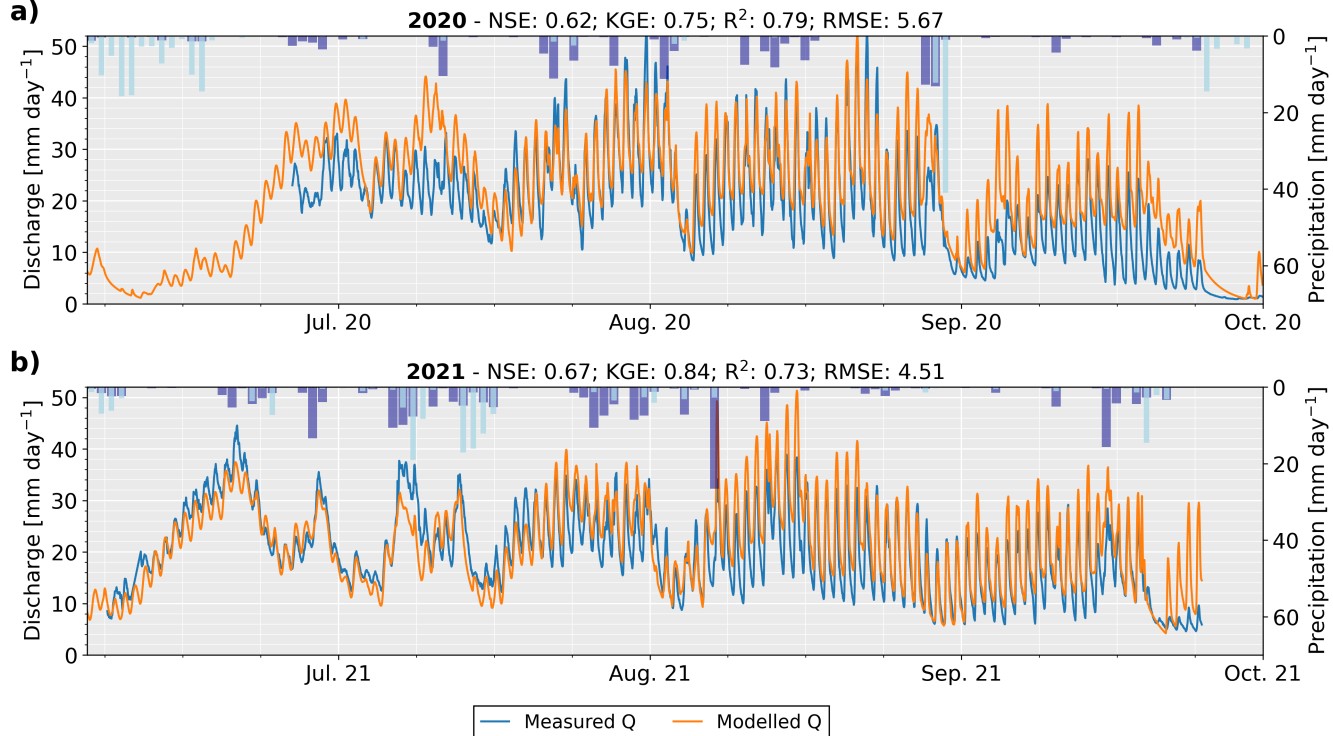

**Figure 6.** Comparison between measured discharge ($Q$) at the glacier portal and modelled discharge from the combined mass balance and transfer model for the melting season of 2020 (a) and 2021 (b). Discharge is expressed in mm per day and corresponds to liters per day divided by the catchment area in m$^2$. For each year, we show the Nash-Sutcliffe efficiency (NSE) and Kling-Gupta efficiency (KGE), as well as the coefficient of determination (R$^2$) and the root mean square error (RMSE). Daily solid (light blue) and liquid (dark blue) precipitations are shown as inverted bars.

cold spells are also well simulated thanks to the slow reservoir. The hydrological transfer model was only calibrated against data from 2020, but the model performance appears as good for 2021. This behaviour is confirmed by the Nash-Sutcliffe efficiency (NSE) and Kling-Gupta efficiency (KGE) criteria (see Gupta et al. (2009) for references) of 0.62 for NSE and 0.67 for KGE in 2020 and 0.75 and 0.84 in 2021 (Fig. 6).

## 4.4 Air temperature and relationship to isotopic composition of precipitation

The relationship between air temperature and precipitation $\delta^2$H appears to be linear with a coefficient of determination (R$^2$) of 0.85 for the mean regression (red line in Fig. 7). However, most of our samples cover the summer season, thus the linear trend is strongly influenced by the limited number of winter precipitation samples. We therefore also highlight the uncertainty margin which was then used to assess the sensitivity of the modelled snow and stream isotopic behavior to this relationship (see Sect. 5.2).



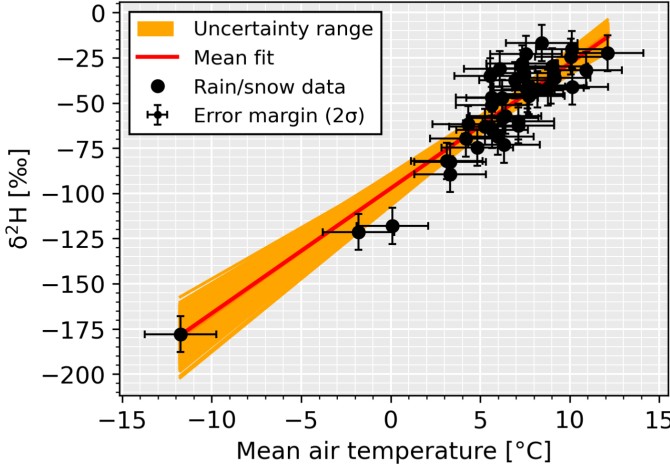

**Figure 7.** Relationship between air temperature measured at the weather station (Fig. 1) and the isotopic composition of 39 precipitation events between 2019 and 2021. For each point we defined a normally distributed error margin of two standard deviations ($2\,\sigma$). The orange area represents the 5000 linear regressions obtained by randomly picking a set of values in the error margin of all points. The red line corresponds to the mean regression.

## 4.5 Isotopic model results

Based on the mean $\delta^2$H of supraglacial ice melt samples, the ice melt $\delta^2$H was set to a fixed value of -109 ‰ which also
reflected the minimum stream $\delta^2$H in late summer, when snow cover is lowest. The snow melt $\delta^2$H was calibrated manually
(Sect. 3.2.1). Parameter $f_{\text{frac,sm}}$ was set to 16 ‰ for $\delta^2$H but had little impact on the results as also shown by Ala-aho et al.
(2017). The sublimation parameter $f_{\text{frac,E}}$ was set to 8 ‰. This represents a rather low fractionation compared to Ala-aho et al.
(2017). $f_{\text{ROS}}$ was manually calibrated to 1 when SWE was below 200 mm and increased linearly until a SWE value of 2000 mm
was reached, then remained constant. This emulates that a thicker snowpack (>2000 mm w.e.), which usually underwent less
melt in summer, only partially mixes with the rain, leading to a smaller increase in $\delta^2$H during ROS events than for shallower
snowpacks for which the snowpack $\delta^2$H mixes more with the rain $\delta^2$H input.

The resulting snow melt $\delta^2$H is shown in Fig. 8a. Snow melt $\delta^2$H is slightly lower than the measured stream $\delta^2$H in the early
melting season, increases during the melting season due to fractionation and ROS events and, as a result, becomes isotopically
less depleted than ice melt. This enrichment is faster in 2021 than 2020 due to much more precipitation and ROS amounts. In
2020, the modelled stream $\delta^2$H appears to fit well with the stream observations, although daily $\delta^2$H variations in July are not
marked in the model. In early July 2021, stream $\delta^2$H is rapidly overestimated by the model, likely due to the complexity of the
ROS processes. In the second part of August 2021, when less rain occurred, stream $\delta^2$H appears better represented.





**Figure 8.** Results of the isotopic model. **(a)** Measured and modelled stream $\delta^2$H at the glacier portal as well as constant $\delta^2$H value assumed for the ice melt composition and the modelled evolution of the snow melt $\delta^2$H . Daily solid (light blue) and liquid (dark blue) precipitations are shown as inverted bars. **(b)** Estimated mixing ratios between ice melt, snow melt, rain and ROS based on the measured stream $\delta^2$H and the modelled $\delta^2$H of the water sources. The shares of rain and ROS were estimated by the transfer model. The black dots indicate the dates of each stream water sample used to estimate the mixing ratios. Grey areas represent periods when no samples were available for more than a day. **(c)** Mixing ratios estimated from the combined mass balance and transfer model only.

## 4.6 Estimation of mixing ratios

We compare the mixing ratios between the four different water sources (rain, ROS, snow melt and ice melt), either estimated based on the simulated discharge of each source (using the mass balance and transfer model) or based on the modelled isotopic compositions of the water sources. As detailed in Sect. 3.4, since we only use water isotopes as tracer, only two components





can be separated (snow and ice melt), while we use the results of the mass balance and transfer model to estimate the water fractions of rain and ROS. The results of the mass balance model (Fig. 8c) show a gradual transition from a snow-dominated discharge towards more ice melt in the late melting season. The estimated contributions of rain and ROS remain usually below

20 %, except for large events (>15 mm d$^{-1}$), during which the peak contribution reaches up to 50 %. The results of the mixing model based on isotopes (Fig. 8b) are more variable. For both years, mixing ratios for the early and late melting seasons are in a similar range as those calculated from the mass balance model. In the middle of the melting seasons, the estimated ratios of snow and ice melt appear much more variable and difficult to interpret.

## 5   Discussion

### 5.1   Mass balance model limitations

We created a simple mass balance model relying on readily available point-based data (precipitation, air temperature, incoming solar radiation). Catchment-wide spatio-temporal variations in temperature and precipitation was modelled using seasonal elevation lapse rates while incoming solar radiation was adapted using a high-resolution DEM to account for slope and aspect. The spatial extrapolation of the meteorological input data relies on an effective calibration procedure that allows the model to

be applied to other locations. Some key aspects of the model are discussed hereafter.

The calibrated seasonal temperature lapse rates showed steeper gradients in summer than winter (Fig. D1c), similar to other studies (e.g., Rolland, 2003; Marshall et al., 2007). At high elevation, the colder summer temperatures obtained with a varying lapse rate compared to a constant lapse rate led to less melt, which in turn influenced the calibration of the precipitation lapse rate. Precipitation lapse rate decreased indeed from about 10 % per 100 m with a constant temperature lapse rate to

about 2 % with a varying lapse rate, which is closer to observations reported over other glaciers (Schaefli et al., 2011). The snow loss and redistribution function on steep slopes, combined with the radiation correction function based on slope and aspect, was essential to correctly represent the timing of the presence/absence of snow on north- and south-facing slopes. Snow sublimation and deposition was also modelled. The amount of snow mass loss due to sublimation remained small compared to snow melt (<5 %). Modelling this process also required additional meteorological data such as wind speed, relative humidity

and surface snow temperature. While we proposed a simple procedure to estimate snow surface temperature, wind speed may vary spatially due to katabatic winds in particular (Greuell and Böhm, 1998; Shaw et al., 2024). Finally, we lack data other than SWE to efficiently calibrate the sublimation function which snow losses may become wrongly attributed to snow melt losses. Snowpack d-excess may provide an interesting additional source of information as it is mostly sensitive to sublimation. Snow sublimation remains uncertain and was here mainly required to estimate the isotopic fractionation in the snowpack rather than

to improve the mass balance model.

Some melt processes were not included in our model. For instance, it does not modify melt over debris-covered glacier areas because they remain limited on Otemma glacier (∼12 %, Linsbauer et al. (2021)). Moreover, our model does not include potential snow melt (Mazurkiewicz et al., 2008) or ice melt (Saberi et al., 2019) caused by warm rain events because their contribution to melt is comparatively very small (Mazurkiewicz et al., 2008; Pomeroy et al., 2016). Residual melt in winter is



also not accounted for in our model, although we measured a winter stream discharge at the glacier portal reaching a minimum of about 0.24 mm d$^{-1}$ (Fig. B1). This residual streamflow is probably due to two main causes. Basal melt in winter could provide such limited flow, creating a thin water film (Flowers and Clarke, 2002), slowly draining through subglacial till or at the contact with bedrock. Alternatively, groundwater contribution from a deeper aquifer could provide such baseflow (see Sect. 5.4). As illustrated in Fig. D2c & d, seasonal discharge obtained from our mass balance model matches very well with

the observed discharge. This seems to further support the assumption that groundwater, basal melt or rain-induced snow melt represent only a marginal water input in summer.

High-resolution daily satellite images from Team Planet (2017) allowed for generation of approximately weekly cloud-free snow maps. Including these snow cover maps as calibration objective function constrained the calibration parameters of our model on steep slopes where no SWE measurements are available, leading to a better representation of the spatial processes.

For example, for 2020 with very limited in-situ measured SWE data, we put 5 times more weight on the snow cover objective function than on the SWE objective function. Compared to having a similar weight between both objective functions, this led to modelled mass losses closer to measured discharge even if it increased the error on measured SWE (Fig. E1c). It appears therefore that snow cover maps may be highly valuable for mass balance model calibration if limited SWE data are available.

### 5.2 Isotopic model limitations and model sensitivity

#### 5.2.1 Ice melt isotopic composition

In this work, we chose a constant ice melt $\delta^2$H composition. Different spatio-temporal studies on ice melt isotopes show conflicting temporal results, with ice becoming either enriched (Penna et al., 2017), depleted (Schmieder et al., 2018) or showing no trends in $\delta^2$H (Maurya et al., 2011). For the Swiss Alps, Jenk et al. (2009) analyzed a 80 m deep ice core and showed some $\delta^2$H variations but no particular trends with depth, except for a shift at the glacier bed. This suggests that older,

deeper ice does not have a significantly different isotopic composition. This may also be true for the surface ice melt, as also supported by Fig. 5, where no change in surface ice melt $\delta^2$H with elevation can be observed. Therefore, using a constant seasonal ice melt $\delta^2$H , equivalent to the mean $\delta^2$H of surface ice melt samples, is considered reasonable for alpine temperate glaciers. Alternatively, using the end-of-summer stream $\delta^2$H at the glacier portal to approximate the ice melt $\delta^2$H composition led to a similar value, although residual snow cover at high elevation may still contribute to discharge as illustrated in our case

by the 20 % snow contribution in September 2020 before the first snowfall (Fig. 8).

#### 5.2.2 Stream isotopic sensitivity to precipitation $\delta^2$H

Several assumptions were made to model the isotopic composition of the snowpack and snow melt, which in turn strongly influenced the modelled stream $\delta^2$H composition. In Fig. 9, we performed a sensitivity analysis by modifying key model parameters to assess their impact on the modelled stream $\delta^2$H .

The largest uncertainty in the modelled stream $\delta^2$H results from the relationship between air temperature and precipitation $\delta^2$H (red area in Fig. 9). Indeed, a change in slope in their linear regression strongly impacts the modelled snowpack $\delta^2$H at



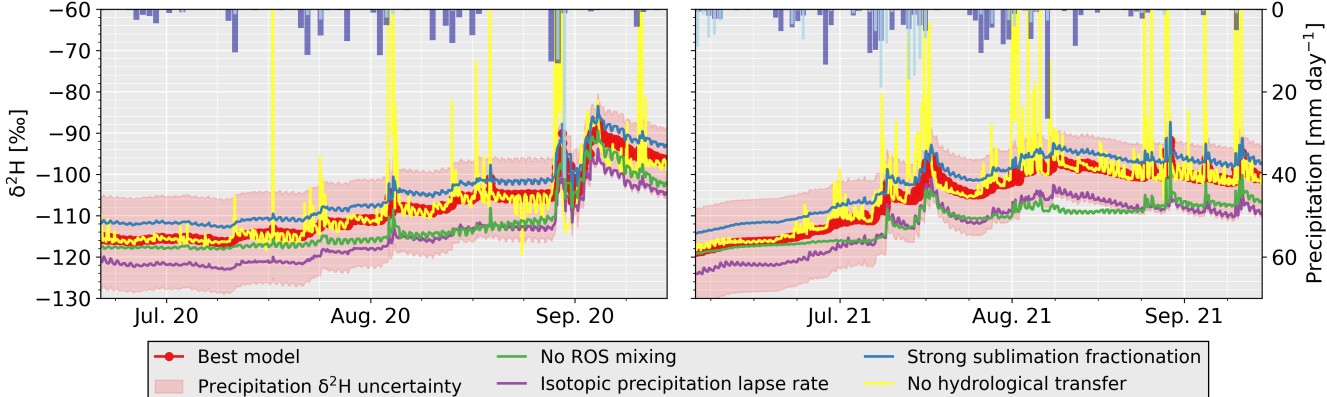

**Figure 9.** Sensitivity analysis of the modelled stream $\delta^2$H at the glacier portal over the melting seasons 2020 and 2021. The curves shown are: Best calibrated model (Best model); uncertainty margin from the relationship between precipitation $\delta^2$H and air temperature (see Fig. 7); model without ROS $\delta^2$H mixing in the snowpack (No ROS mixing, $f_{ROS} = 0$); model with an isotopic precipitation $\delta^2$H lapse rate of -1 ‰ per 100 m; model with a strong isotopic sublimation fractionation ($f_{frac,E} = 80$ ‰); model without hydrological transfer (No transfer). Daily solid (light blue) and liquid (dark blue) precipitations are shown as inverted bars.

peak snow accumulation, which in turn strongly modifies the stream $\delta^2$H at the onset of snow melt. Moreover, precipitation $\delta^2$H may be influenced by other parameters than air temperature. In this study, the limited number of bulk snowpack $\delta^2$H samples limits the calibration/evaluation of this approach. Nonetheless, at least one recent study providing a much more in-

depth analysis of snowpack $\delta^2$H profiles validated the strong relationship between the snowpack $\delta^2$H and air temperature (Carroll et al., 2022a).

In addition, precipitation $\delta^2$H may also vary with elevation. No isotopic precipitation lapse rate was used in this work, as it could not be observed. This lack of $\delta^2$H lapse rate may be due to the complex air flow above a high elevation terrain, where air parcels may stagnate or flow down-valley (Schäppi, 2013). This modifies vapor condensation and thus invalidates

the relationship between elevation and a depletion of heavy isotopes in the water vapor (Galewsky, 2009). Such an absence of $\delta^2$H lapse rate trend was also observed from high elevation precipitation data in Switzerland, especially in winter (Kern et al., 2014). Nonetheless, at least one recent study provided a detailed description of multiple snow profiles with elevation (Carroll et al., 2022a). While they measured a $\delta^2$H precipitation lapse rate of -1.28 ‰ per 100 m, they show no statistically significant differences with elevation in the snowpack bulk isotopic composition at the time of maximum end-of-winter snow

accumulation. They attribute this to the persistence of warm, enriched early winter snow at high elevation and different rates of snow accumulation and sublimation in winter. In our model, including a $\delta^2$H lapse rate of -1 ‰ per 100 m leads to a more depleted snowpack before the onset of snowmelt (purple curve in Fig. 9). It also reduces the increase of stream $\delta^2$H in summer due to ROS events. A winter $\delta^2$H lapse rate may lead to a too depleted snowpack compared to observed (purple curve in Fig. 9). However, a summer $\delta^2$H lapse rate may be required to better represent the stream $\delta^2$H during wet summers such as

2021, where the effect of ROS (without $\delta^2$H lapse rate) led to a too rapidly enriched modelled stream $\delta^2$H (Fig. 8a).





### 5.2.3 Stream isotopic sensitivity to ROS

ROS is also influenced by the summer precipitation lapse rate which is assumed to be similar to winter snowfall (increase of 2 % per 100 m). Similar to the $\delta^2$H lapse rate, precipitation lapse rates may become flat or even negative in the Swiss Alps in summer above 2500 m a.s.l. (e.g., Schäppi, 2013). It is therefore likely that precipitation shows a smaller increase with

elevation in summer than in winter, resulting in an overestimation of the summer precipitation amounts when using a fixed annual lapse rate. This impacts how much ROS mixes within the snowpack and how it modifies the snowpack $\delta^2$H . Both precipitation amounts and $\delta^2$H precipitation lapse rates appear therefore challenging to accurately model, which may lead to large uncertainties even for relatively dry years such as 2020.

    In addition, how ROS mixes, refreezes or leaks through the snowpack remains a major question. Neglecting ROS $\delta^2$H

mixing in the snowpack ($f_{\text{ROS}} = 0$) leads to a much slower enrichment of the snowpack in heavy isotopes during the melting season. This process appears to be the main driver of the seasonal increase in the stream $\delta^2$H value (green curve in Fig. 9). Only limited experimental work exists on this topic. For instance, for an in-situ ROS experiment, Juras et al. (2017) showed preferential flow of rain water in non-ripe cold snow and limited time for isotopic exchange with the snowpack. For a ripe snowpack, they showed that less than 50 % of rain water was directly released from the snowpack, leading to partial mixing of

the rain water with the snowpack. In another in-situ study, Rücker et al. (2019a) showed that interactions between rain water and the snowpack were mostly influenced by the residence time of the rain water in the snowpack, which mostly depended on snow depth and rainfall amounts. In this work, we attempted to provide a simple formula for the mixing of ROS in the snowpack based on the SWE amounts (see Sect. 3.2.3), and the best calibration was achieved with a complete incorporation of ROS in the snowpack when the snowpack was thin and likely isothermal (<200 mm w.e.) and a partial mixing of 50 % when

the snowpack was thick (>2000 mm w.e.). The validity of such a simple approach should, however, be explored in further work.

### 5.2.4 Stream isotopic sensitivity to snow fractionation

Based on our calibration, we used a rather low sublimation fractionation factor ($f_{\text{frac,E}} = 8 ‰$) compared to Ala-aho et al. (2017), so that the differences with no sublimation are limited. Using a ten times higher sublimation fractionation factor (blue

curve in Fig. 9) leads to a more enriched snowpack before July but does not particularly impact the stream $\delta^2$H evolution in summer, which conserves a winter $\delta^2$H offset compared to the best model. This is due to the fact that the modelled sublimation mostly occurs during spring when cold dry air has a lower vapor density than the snow surface. We also estimated a small deposition amount (34 and 14 mm w.e. for 2020 and 2021), but deposition was not included in the isotopic module as the isotopic composition of the water vapor in the air appears very complex as discussed above for precipitation. Its effect on snow

fractionation remains therefore unclear.

    Similar to Ala-aho et al. (2017), the parameter governing liquid fractionation of the snowpack ($f_{\text{frac,sm}}$, not shown in Fig. 9) had only limited influence on the modelled summer stream $\delta^2$H . Nonetheless, some studies (e.g., Taylor et al., 2001) have reported liquid fractionation during melt as an important driver of isotopic enrichment. For this work, however, this process



defined by Eq. 18 seems to play a minor role due to the rapid increase in the number of melt days ($d_{melt}$). While we retained
the original equation from Ala-aho et al. (2017), its validity could be further explored.

### 5.2.5   Stream isotopic sensitivity to snow melt transfer

Finally, we show the isotopic composition of snow melt if we simply take the average value of all snow melt grid cells without
the transfer module (yellow curve in Fig. 9). In this case, the signal shows more variability and small peaks mainly due to the
effect of summer snowfall or ROS events. The snow transfer to the outlet contributes to smoothing out short term variations in
$\delta^2$H , while the signal remains similar when no precipitation occurs.

### 5.3   Limitations of the hydrological transfer model

We modelled water transfer from its point of mobilisation to the catchment outlet based on an estimation of the mean transit
time of the water through four different landcover types using gamma distributions. While this approach is somewhat different
from more widely used bucket-type approaches (e.g., Schaefli et al., 2005), it leads to relatively similar results, as the spread
of the gamma distribution can be related to a storage recession time constant. This approach assumes that the water transfer
is driven by advection but does not depend on previous conditions such as antecedent wetness or the amount of storage in a
compartment (snowpack, groundwater or en-/subglacial system). It was, however, largely shown in the hydrological literature
that older water tends to be rapidly and preferentially released to the stream during rain events, suggesting that "new" water
inputs in a catchment tend to activate and push "older" pre-event water out of soils and groundwater reservoirs (e.g., McDonnell,
1990; Kirchner, 2003). Such mechanisms are usually not well represented in hydrological transfer models but can be assessed
using water isotopes.

     In Fig. 10, we highlighted three periods during the early, mid and late melting season during which rain events occurred. In
the upper plots, discharge response to rain events appears fast (within an hour) during all periods and streamflow seems only
shortly influenced by rain events (during about a day). However, when looking at the stream isotopic response (second row of
plots in Fig. 10), it appears that the stream $\delta^2$H seems to be longer influenced by the rain (higher $\delta^2$H value) before it goes
back to its pre-event composition. This is especially visible for the second half of July 2020 (Fig. 10a). There, the measured
stream $\delta^2$H appears to increase more gradually than discharge in response to rain events and only return to the baseline value
about 5 days later. This highlights the typical old water paradox mentioned before, for which hydrograph response is swift
but the water is composed of more pre-event water (Kirchner, 2003). Finally, in the lower panel of Fig. 10a, we show that our
hydrologic transfer model fails to correctly represent the fraction of rain event water.

     Later in the summer, in mid-August 2020 (Fig. 10b), the response to rain events seems better represented by our model.
Finally, during September 2021 (Fig. 10c), the isotopic signal shows a fast decrease within about one day, with a first peak
rapidly after the rain event and a second peak about 6 hours later, while no increase in discharge is observed. This second peak
seems to come from delayed snow melt, given that d-excess during this peak significantly decreased to 5 ‰, while it returns to
its baseline value of 10 ‰ within the next 3 hours, synchronous with the end of the isotope peak (not shown here).





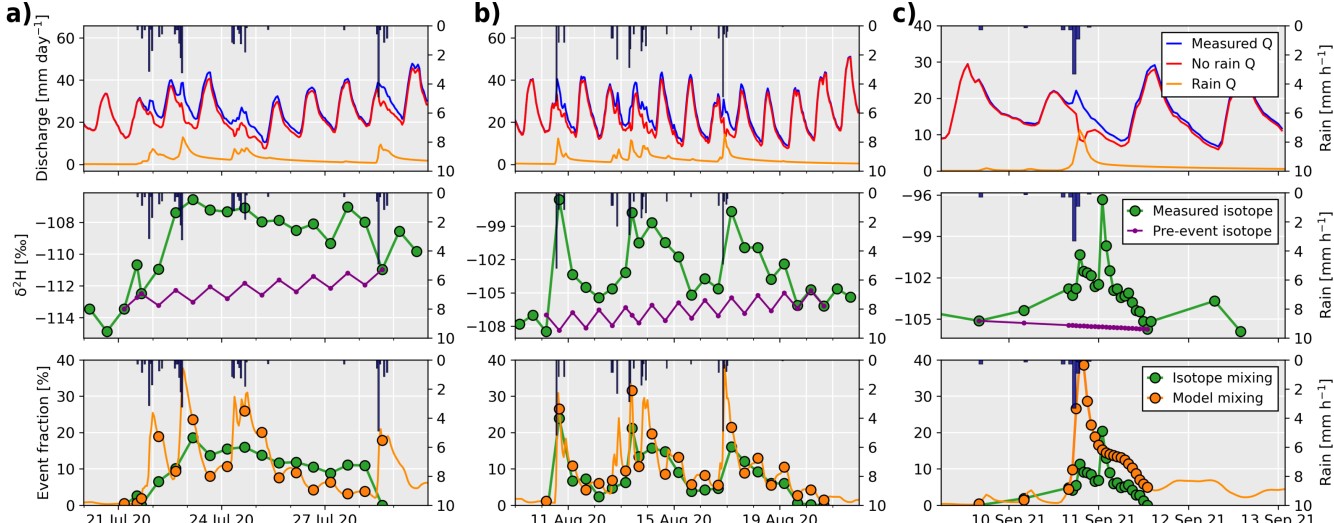

**Figure 10.** Comparison of the estimated fraction of the total discharge originating from rain events during the early 2020 **(a)**, mid 2020 **(b)** and late 2021 melting season **(c)**. The upper figures show the measured discharge (blue curve) at the glacier portal and the sum of rain and ROS discharge (event discharge) estimated from the model (orange). The red curve shows the measured discharge without the modelled event water. The central figures show the measured stream $\delta^2$H (green) and an estimation of the pre-event $\delta^2$H based on a simple linear interpolation between the pre- and post-event stream compositions. If pre-event stream $\delta^2$H showed small daily variations (due to the increase of ice melt during the day), this behavior was added to the linear interpolation to better represent the baseline $\delta^2$H composition. Lower figures show the fraction of event water (rain and ROS) in the streamflow estimated either based on the modelled event discharge (see upper figures) or based on isotopes (see central figures).

The slower release of event water occurs mainly during the early season and is faster in August. This suggests that this phenomenon is linked to a seasonal water storage which decreases over the season. We hypothesise that this storage could be either linked to the seasonal switch from an inefficient distributed en-/subglacial drainage system to a more efficient channelized system or to the gradual disappearance of the snow cover. In our modelling framework, we implemented a fast and slow storage which collects the water before reaching the glacier outlet (see Sect. 3.3.5). Nonetheless, it appears that using a fully mixed storage was not adequate and that this storage should allow the release of older water to represent the isotopic signal more accurately.

To better represent this mechanism, we also attempted to calibrate the hydrologic transfer parameters with and without including stream $\delta^2$H as an objective function. Overall, discharge NSE and KGE were similar for both calibrations. Parameter estimation was not particularly different except for hillslope parameters, which led to a slower transmission of water with a slower decrease in discharge when isotopes were used for calibration. This results in a more dampened stream isotopic response during rain events, closer to the measured stream $\delta^2$H . Including isotopes for calibration modifies therefore the internal mechanisms of the hydrological transfer model, but it remains unclear if such changes reflect more realistic processes or a simple trade-off due to the inability of the model structure to represent the preferential release of older water.





### 5.4 The role of groundwater

The contribution from groundwater sources was not included in our modelling approach. To some extent, delayed lateral subsurface flow (Carroll et al., 2019) from elevated snow melt transmitted through the hillslopes is estimated by the hillslope and snow routing modules, but bedrock exfiltration was not considered. Groundwater contributions from the bedrock may not be completely negligible as the discussion of groundwater storage has recently shown for Swiss alpine glaciers (Müller et al., 2022; Oestreicher et al., 2021). For our catchment, it is possible that a part of the early snow melt contributes to recharge the highly fractured bedrock and is then redistributed towards the late melting season when snow cover is limited. In Otemma, a winter dynamic bedrock storage was estimated to be equivalent to 40 mm of water stored over the entire catchment (Müller et al., 2022). This seasonal storage may increase in spring to about 60 to 100 mm as a result of snow melt recharge as suggested for other glacierized catchments (Hood and Hayashi, 2015; Oestreicher et al., 2021). This recharge is potentially visible in Fig. D2d (cumulative discharge from 0 to 500 mm), as the cumulative modelled discharge in the early melting season is about 50 mm larger than measured, which could be due to some snow melt water infiltrating into the bedrock and not being routed to the glacier portal. Later in the melting season, groundwater bedrock drainage may then lead to higher modelled discharge than measured (Fig. D2d, cumulative discharge from 1000 to 1500 mm). However, this amount of storage release lies in the same range as the RMSE of the differences between observed and modelled SWE in 2021 (Fig. E1b), although the mean error is close to 0. Therefore, the amount of groundwater storage in the bedrock remains in the statistical error margin of the model.

Stream EC was always higher than ice and snow melt EC (Fig. B1), and largely increased in winter, which also points to the potential contribution of a groundwater reservoir. However, as highlighted in some studies (e.g., Sharp et al., 1995; Hindshaw et al., 2011), subglacial weathering at the contact with the bedrock or sediments leads to an increase in solutes in the meltwater. Since EC is not a conservative tracer, groundwater estimation via EC measurements may lead to much larger uncertainty than what some studies may suggest, as subglacial weathering cannot be easily quantified. In any case, groundwater contribution should not largely impact the stream $\delta^2$H, as the $\delta^2$H of bedrock leakages was found to be close to ice melt $\delta^2$H (Fig. 3).

### 5.5 Towards a better estimation of the snow isotopic composition

From the discussion above, it appears that reconstructing snowpack $\delta^2$H based on meteorological data and a mass balance model is possible but large uncertainties may arise from the definition of parameters as shown in Fig. 9. We suggest that bulk snow samples estimating the average $\delta^2$H of the snowpack are one prerequisite to evaluate this approach. At the time of maximum end-of-winter snow accumulation during four years, Carroll et al. (2022a) showed relatively limited bulk $\delta^2$H differences for about 13 repeated snowpack profiles. This suggests that a limited number of bulk snow samples may be sufficient to calibrate or evaluate the winter isotopic model. Ideally, simple snow lysimeters could also be installed (Rücker et al., 2019b).

The snowpack $\delta^2$H evolution during the melting season remains more challenging to model, mainly because of the complex processes involved during ROS events. In our case, we relied on high-resolution stream $\delta^2$H data, which helped to constrain the snow melt evolution by setting simple rules: snow melt $\delta^2$H should always be lower than stream $\delta^2$H in the early melting season if there is strong evidence that snow is the more depleted end-member. Later in the melting season, stream $\delta^2$H was less




useful as snow melt $\delta^2$H was more enriched than ice melt $\delta^2$H . Additional bulk snow samples could help to better calibrate the model, but snowpack sampling at high elevation may be compromised by difficult access.

Based on the results of the hydrological transfer module, the delay between snow melt and its arrival at the catchment outlet was usually less than a day. As a result, the snow melt $\delta^2$H at the catchment outlet modelled using the calibrated transfer scheme was close to the $\delta^2$H value calculated using the simple daily melt amount-weighted mean snow melt $\delta^2$H of all grid cells (Fig. 9). This suggests that the hydrological transfer may not be necessary to estimate the daily snow melt $\delta^2$H value at the catchment outlet. During rain events, the hydrological transfer allows improved smoothing of the signal, but relatively similar

results could likely be obtained by applying a low-pass filter to the weighted mean snow melt $\delta^2$H data.

Therefore, it seems possible to estimate the temporal evolution of the outlet snow melt $\delta^2$H based solely on mass balance modelling (with necessary meteorological data) and snow observations, without relying on discharge data, which are required to calibrate a full hydrological transfer model and are usually difficult to acquire for high-elevated and more remote catchments.

Finally, incorporating d-excess in the modelling framework as proposed by Carroll et al. (2022b) may provide additional

information concerning the rate of sublimation in the snowpack and contribute to better calibrate the model. D-excess may also be used as an additional tracer along $\delta^2$H for mixing models to separate snow melt from ice melt as proposed by Sprenger et al. (2023). However, processes of sublimation and deposition affecting the snowpack $\delta^2$H are complex (Lambán et al., 2015) and remain challenging to model precisely.

### 5.6    Mixing model limitations

In this research, we have proposed a way to better characterize the temporal evolution of snow melt $\delta^2$H in highly glacierized catchments. This method, especially if validated with more snowpack or snow melt observations, should contribute to limit snow melt isotopic uncertainties due to the spatio-temporal variability of the snowpack $\delta^2$H. Nonetheless, even with such an approach, answering the question of water sources mixing ratios at the catchment outlet based on isotopes (Fig. 8b) appears very challenging. As discussed in Sect. 5.3, during rain and ROS events, the transfer of water to the stream may be complex,

with a preferential release of older water (from the snow or from the hillslopes), which may not be captured by our model, and may strongly influence the estimated mixing ratios. In Fig. 11, we provide an analysis of the ice and snow melt shares when we estimated a rain water fraction of less than 5 %. Even in that case, water sources mixing results based on isotopes (Fig. 11a & b) appear very variable, especially during the mid melting season. It appears therefore that during the mid melting season, snow melt $\delta^2$H increases and reaches values close to ice melt $\delta^2$H , so that even a slight error in the estimation of their $\delta^2$H

values leads to a large uncertainty in their estimated shares using a mixing model approach.

Finally, no significant mixing differences can be observed with (Fig. 11b) or without hydrological transfer (Fig. 11a) for snow melt, which illustrates again that snow melt transfer may not be necessary to estimate the snow melt $\delta^2$H in small highly glacierized catchments.

We conclude that separating the shares of ice and snow melt in the glacier meltwater stream with isotopes remains a very

difficult task, and a simple mass balance approach likely leads to better results on a weekly scale. Nonetheless, the stream $\delta^2$H signal provides interesting insights in some of the internal mechanisms which modulate the glacio-hydrological response on



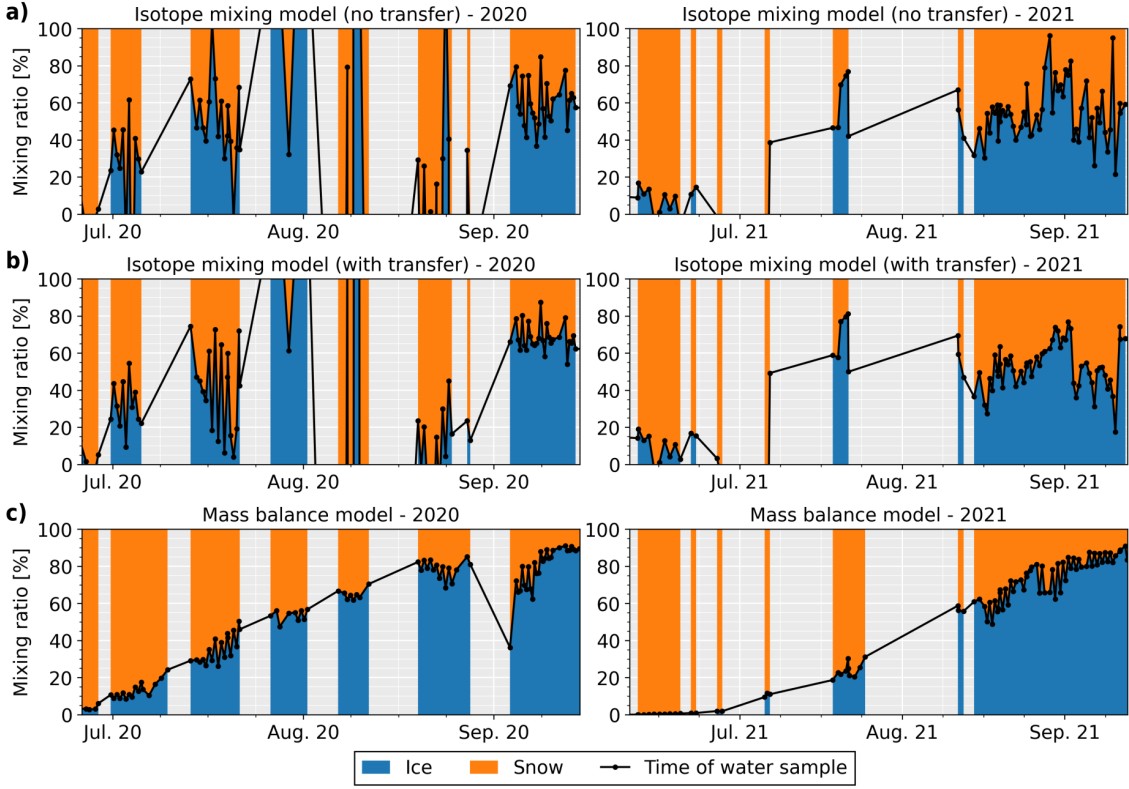

**Figure 11.** Mixing ratio between snow and ice melt when rain contribution to streamflow is less than 5 %. **(a)** Results based on snow melt $\delta^2$H estimated by a simple weighted-average of all model grid cells (without transfer). **(b)** Results when snow melt $\delta^2$H is routed to the glacier terminus. **(c)** Results from the mass balance and transfer model without isotopes. Black points show dates when samples were taken.

a sub-daily scale, and may therefore contribute to develop more sound physically-based models. Outside of the main melting season, discharge becomes lower and other marginal sources of water, such as groundwater, may become relatively more important and should not be neglected.

## 6 Conclusions

The aim of this research was to (i) provide a framework for extrapolating point-based water stable isotope data to an entire highly glaciated catchment, (ii) model their signal transfer to the catchment outlet, and (iii) assess how water isotopes can be used to estimate the proportions of different water sources at the outlet.

In glacierized catchment, the use of water isotopes remains challenging as the isotopic composition of the water sources shows strong spatio-temporal variability and their composition may partially overlap, leading to large uncertainties in the use of mixing models (e.g., Schmieder et al., 2016, 2018; Zuecco et al., 2019; Penna et al., 2017).



Our field data highlighted the large isotopic spatial variability of the surface snow samples which showed no particular trend with elevation and completely different values than the snowpack at 10 to 20 cm. This suggests that only bulk samples of the entire snowpack should be sampled to represent the snowpack. As such isotopic data are difficult to obtain in the field and may

evolve over time, we proposed to estimate the catchment-wide snowpack $\delta^2$H based on a mass balance approach coupled to a snow isotopic module based on the previous work of Ala-aho et al. (2017).

This modelling framework aims to limit the collection of snowpack data over the season by using more readily available meteorological observations but still requires a number of SWE and ice ablation data for calibration. When limited SWE and ice ablation data are available, we showed the benefit of using an automatic optimisation algorithm such as PEST-HP

to better constrain model parameters, and also showed how high spatio-temporal satellite products contributed to an efficient model calibration. Seasonal streamflow volumes agreed well with the melt contribution estimated by the mass balance model, and discharge data was only required to calibrate an hourly water transfer module accounting for hillslope, snow and glacier routing.

To correctly model the snowpack $\delta^2$H during the accumulation period, we used on a regression curve between precipitation

$\delta^2$H and air temperature. Although other factors, such as the origin of the moisture source, may influence the precipitation $\delta^2$H , the relationship with air temperature was significant and sufficient to reproduce the snowpack $\delta^2$H . Here, the use of bulk snow $\delta^2$H samples is recommended to validate the snowpack $\delta^2$H around the period of maximum end-of-winter snow accumulation. The resulting modelled stream $\delta^2$H at the catchment outlet agreed well with observed streamflow during the melt season, except for periods when ROS was important. How ROS is integrated and released from the snowpack during the melt season

also appears to be difficult to model, affecting the composition of streamflow $\delta^2$H and thus limiting the use of mixing models during ROS events. Although stream $\delta^2$H is modelled correctly in absence of ROS, the results of a mixing model solely based on the modelled isotopic composition of rain, snow and ice melt led to unrealistic results for most of the melt season. In fact, for highly glacierized catchments, the gradual snow $\delta^2$H enrichment during the melt season leads to a snow melt $\delta^2$H signal close to ice melt $\delta^2$H , which in turn makes hydrograph separation based solely on isotopes very challenging and therefore not

advisable.

Finally, our glacio-hydrological model combined with streamflow $\delta^2$H may provide interesting insights into the physical mechanisms of water release and transfer to the outlet. For example, we showed that during rain events, older pre-event water was preferentially released during the early hydrograph response, with a slower release of event water than observed in the hydrograph, suggesting the presence of a temporary storage that diminished during the melt season.

We conclude that mixing models based on water isotope samples in highly glacierized catchment should only be considered where snow and ice melt $\delta^2$H have clearly different values. For more complex glacio-hydrological models, the inclusion of water isotopes may provide additional data to better constrain model parameters but their use to better constrain the shares of snow and ice may be limited. To improve the use of water isotopes in models, more work is needed to characterise the complex processes associated with the precipitation isotope lapse rates, with wind dynamics on glaciers and with ROS incorporation

into the snowpack.



# Appendix A: List of glacio-hydrological model parameters

**Table A1.** Glacio-hydrological model parameters with initial and calibrated values for 2020 and 2021.

| Model parameters | Units | Initial value | Calibration 2020 | Calibration 2021 |
|---|---|---|---|---|
| Mass balance model parameters | | | | |
| Temperature lapse rate ($\mu_{\Delta_T}$) | [days] | 150 | 136.6 | 156.0 |
| Temperature lapse rate ($\sigma_{\Delta_T}$) | [days] | 75 | 77.6 | 58.5 |
| Temperature lapse rate ($f_{\Delta_T,\text{range}}$) | [°C per 100 m] | 0.2 | 0.180 | 0.180 |
| Temperature lapse rate ($f_{\Delta_T,\text{inc}}$) | [°C per 100 m] | 0.35 | 0.417 | 0.484 |
| Precipitation lapse rate ($\Delta_P$) | [% per 100 m] | 2 | 2.16 | 2.63 |
| Snow precipitation factor ($f_{\text{corr,snow}}$) | [-] | 2 | 1.90 | 2.33 |
| Temperature melt threshold ($T_{\text{melt}}$) | [°C] | 1 | 0.97 | 1.28 |
| Temperature factor ($f_{\text{melt,T,snow}}$) | [mm h$^{-1}$ °C$^{-1}$] | 0.13 | 0.127 | 0.132 |
| Shortwave radiation factor ($f_{\text{melt,rad,snow}}$) | [mm m$^2$ h$^{-1}$W$^{-1}$] | 3.50E-03 | 3.40E-03 | 3.49E-03 |
| Temperature factor ($f_{\text{melt,T,ice}}$) | [mm h$^{-1}$ °C$^{-1}$] | 0.3 | 0.307 | 0.301 |
| Shortwave radiation factor ($f_{\text{melt,rad,ice}}$) | [mm m$^2$ h$^{-1}$W$^{-1}$] | 1.00E-03 | 1.32E-03 | 1.50E-03 |
| Sublimation factor ($E_{\text{sp}}$) | [°C] | 8 | 7.0 | 7.4 |
| Slope factor ($f_\theta$) | [-] | 1.25 | 1.27 | 1.41 |
| Slope threshold ($\theta_{\text{redist,thresh}}$) | [°] | 30 | 31.9 | 31.7 |
| Radiation slope factor ($f_{\text{rad,slope}}$) | [-] | 1.5 | 1.71 | 1.41 |
| Radiation slope threshold ($\theta_{\text{max,rad}}$) | [°] | 60 | 67.2 | 59.8 |
| Radiation aspect factor ($\gamma_{\text{max,rad}}$) | [-] | 3 | 2.72 | 1.97 |
| Isotope model parameters | | | | |
| Snowpack melt fractionation factor ($f_{\text{frac,sm}}$) | [‰] | 8 | 16 | 16 |
| Snowpack sublimation fractionation factor ($f_{\text{frac,E}}$) | [‰] | 40 | 8 | 8 |
| Rain on snow incorporation factor ($f_{\text{ROS}}$) | [-] | 1 | 0.5 (SWE>2000 mm) to 1 (SWE<200 mm) | 0.5 (SWE>2000 mm) to 1 (SWE<200 mm) |
| Transfer model parameters | | | | |
| Hillslope dispersion coefficient ($D_{\text{hillslope}}$) | [-] | 1 | 0.5 | 0.5 |
| Hillslope hydraulic conductivity ($K_s$) | [m s$^{-1}$] | 0.05 | 0.01 | 0.01 |
| Channelized system dispersion coef. ($D_{\text{channelized}}$) | [-] | 1 | 7.92 | 7.92 |
| Channelized system velocity ($v_{\text{channelized}}$) | [m s$^{-1}$] | 0.8 | 0.70 | 0.70 |
| Distributed system dispersion coef. ($D_{\text{distributed}}$) | [-] | 0.5 | 1.12 | 1.12 |
| Distributed system velocity ($v_{\text{distributed}}$) | [m s$^{-1}$] | 0.1 | 0.26 | 0.26 |
| Snowpack dispersion coefficient ($D_{\text{sp}}$) | [-] | 1 | 1.47 | 1.47 |
| Snowpack infiltration velocity ($v_{\text{sp}}$) | [mm h$^{-1}$] | 1200 | 4913 | 4913 |
| Rain on snow dispersion coef. ($D_{\text{ROS}}$) | [-] | 1 | 1.41 | 1.41 |
| Rain on snow infiltration velocity ($v_{\text{ROS}}$) | [mm h$^{-1}$] | 1200 | 9079 | 9079 |
| Slow reservoir response time constant ($k_{\text{slow}}$) | [h] | 40 | 51 | 51 |
| Fast reservoir response time constant ($k_{\text{fast}}$) | [h] | 2 | 1 | 1 |
| Slow reservoir fraction ($f_{\text{reservoir}}$) | [-] | 0.5 | 0.45 | 0.45 |





## Appendix B: Stream data

**Figure B1.** Measurements performed from late June 2020 to mid-September 2021 in the glacial stream directly at the glacier portal. **(a)** Estimated discharge data based on stream stage and a discharge rating curve. **(b)** Water electrical conductivity (EC). **(c)** Water stable isotopes ($\delta^2$H) measurements with dots representing the date of the sampling (usually twice a day in summer). **(d)** Corresponding isotopic d-excess. The inverted blue bars show the measured daily rainfall amounts measured at the weather station.



## Appendix C: Stable water isotope measurements of snow and ice over time

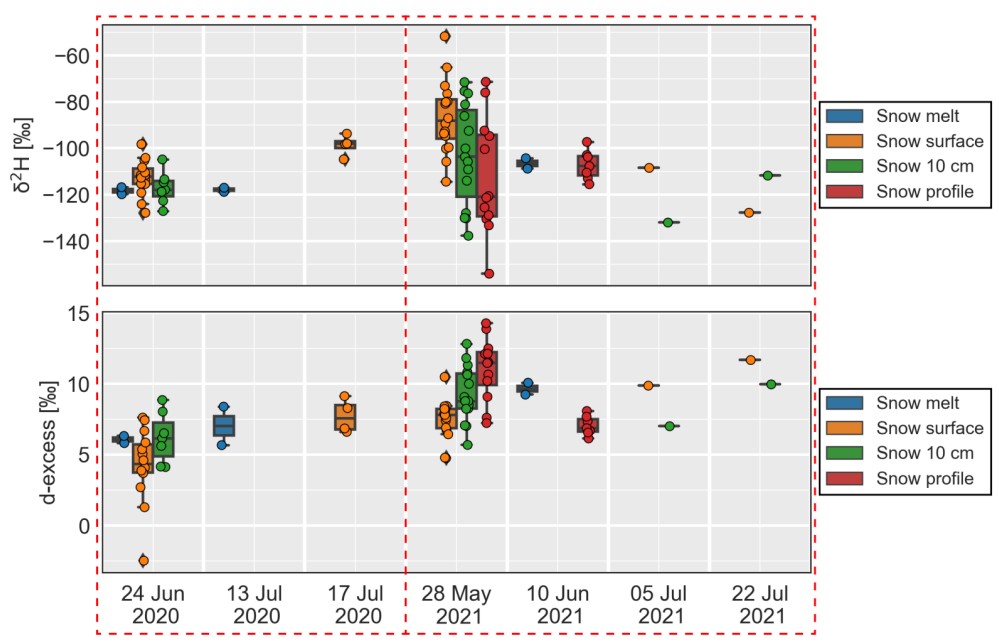

**Figure C1.** Temporal isotopic ($\delta^2$H) and d-excess evolution of snow samples for each sampling dates. Dots show the values distribution. The red dashed squares separate each year of data.

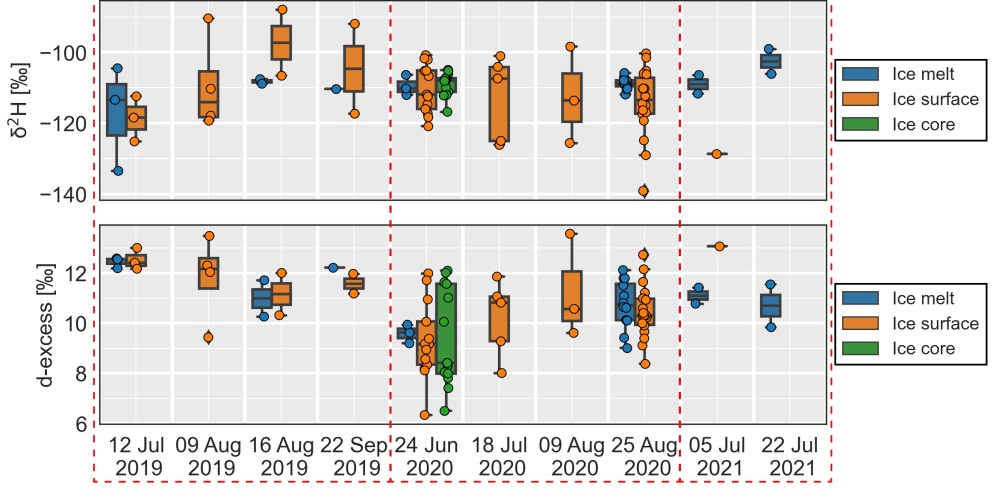

**Figure C2.** Temporal isotopic ($\delta^2$H) and d-excess evolution of ice samples for each sampling dates. Dots show the values distribution. The red dashed squares separate each year of data.



## Appendix D: Calibrated snow mass balance functions

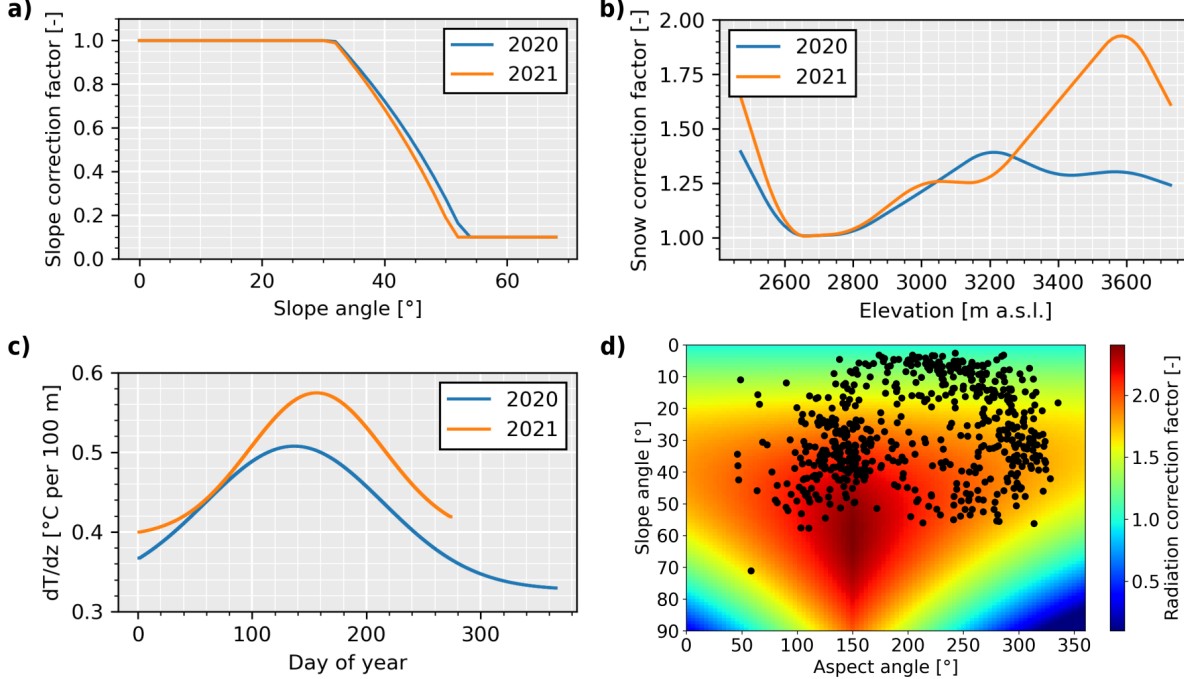

**Figure D1.** Results of the calibrated mass balance functions for 2020 and 2021 derived using PEST-HP. **(a)** Slope correction factor showing where snow reduction occurs (if the terrain slope angle is higher than $\theta_{\text{redist,thresh}}$, with a reduction rate $f_\theta$. **(b)** Corresponding snow correction function ($f_{\text{redist}}$) if slope is smaller than $\theta_{\text{redist,thresh}}$. Snow redistribution is estimated based on a simple relationship with terrain elevation. **(c)** Temperature lapse rate ($\Delta_T$) calibration for both years. **(d)** Radiation correction factor ($f_{\text{rad,slope,tot}}$-$f_{\text{rad,aspect,tot}}$) for 2021 based on terrain slope and aspect. Black dots correspond to all grid cells of the model discretization.





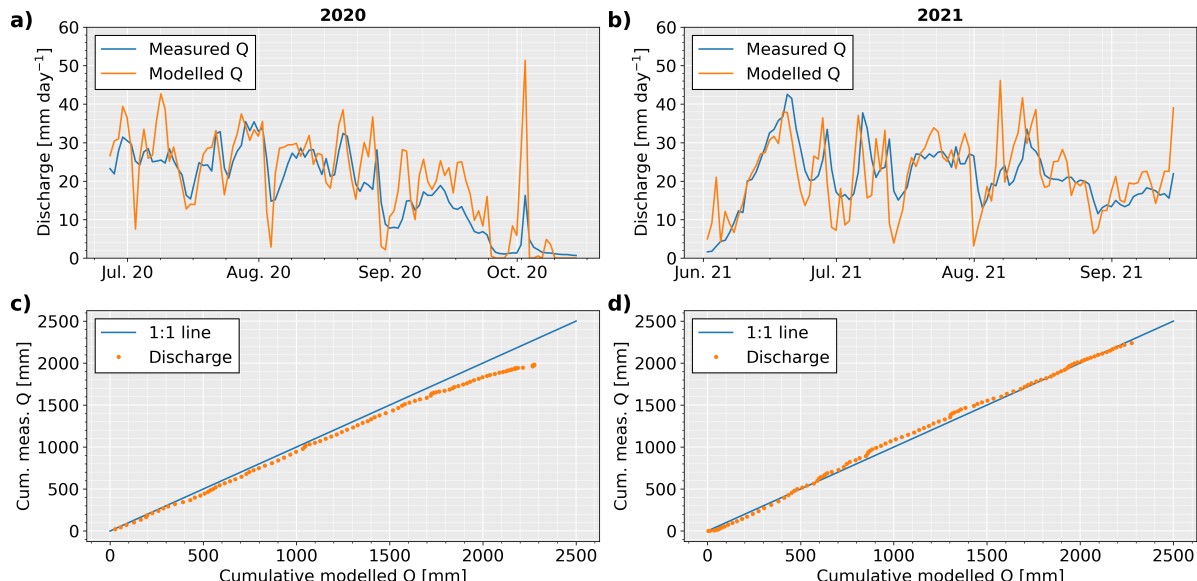

**Figure D2.** Measured and modelled (from rainfall, snow melt, ice melt) discharge (Q) at the catchment outlet for the melting season in 2020 **(a),(c)** and 2021 **(b),(d)**. **(c),(d)** show the comparison of the cumulative total modelled discharge versus measured discharge at the glacier portal.





**Appendix E: Snow and ice mass balance maps**

**Figure E1.** Modelled and measured snow accumulation for 2020 **(a)** and 2021 **(b)** and ice ablation for 2020 **(c)** and 2021 **(d)**. The left figures show the modelled snow accumulation (SWE) or ice ablation for the corresponding year. The middle figures show the measured point mass balances (winter mass balance in spring (**(a)**,**(b)**) and annual mass balance in late summer (**(c)**,**(d)**)). The right figures show the difference between measured and modelled mass balance. The corresponding mean error and root mean square error for each map is also highlighted (all values in w.e.). For the modelled snow accumulation (SWE), 2020 corresponds to the measurement date of 29.06.2020, and 2021 to 28.05.2021. Ice ablation corresponds to the measured annual ablation from 01.10.2019 to 18.09.2020 for 2020 and to the measured annual ablation from 18.09.2020 to 24.09.2021 for 2021.







**Figure E2.** Modelled and mapped snow cover 2020 (part 1). The left figures show the modelled SWE with the corresponding date. The second figure row shows the modelled snow presence (1) or absence (0). The third figure row shows the mapped snow presence (1) or absence (0) based on Planet satellite imagery. The last (right) row shows the mismatch between mapped and modelled snow presence and absence (1 for wrong modelled snow cover presence, -1 for wrong modelled snow cover absence). The figure lines show different dates as indicated on the left row map titles.





**Figure E3.** Modelled and mapped snow cover 2020 (part 2). The left figures show the modelled SWE with the corresponding date. The second figure row shows the modelled snow presence (1) or absence (0). The third figure row shows the mapped snow presence (1) or absence (0) based on Planet satellite imagery. The last (right) row shows the mismatch between mapped and modelled snow presence and absence (1 for wrong modelled snow cover presence, -1 for wrong modelled snow cover absence). The figure lines show different dates as indicated on the left row map titles.







**Figure E4.** Modelled and mapped snow cover 2021 (part 1). The left figures show the modelled SWE with the corresponding date. The second figure row shows the modelled snow presence (1) or absence (0). The third figure row shows the mapped snow presence (1) or absence (0) based on Planet satellite imagery. The last (right) row shows the mismatch between mapped and modelled snow presence and absence (1 for wrong modelled snow cover presence, -1 for wrong modelled snow cover absence). The figure lines show different dates as indicated on the left row map titles.




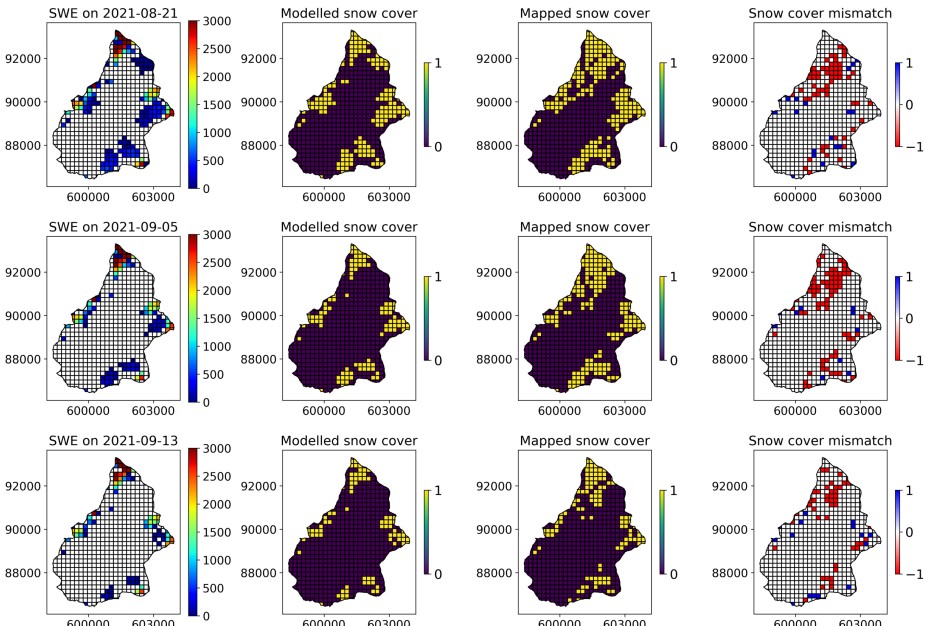

**Figure E5.** Modelled and mapped snow cover 2021 (part 2). The left figures show the modelled SWE with the corresponding date. The second figure row shows the modelled snow presence (1) or absence (0). The third figure row shows the mapped snow presence (1) or absence (0) based on Planet satellite imagery. The last (right) row shows the mismatch between mapped and modelled snow presence and absence (1 for wrong modelled snow cover presence, -1 for wrong modelled snow cover absence). The figure lines show different dates as indicated on the left row map titles.

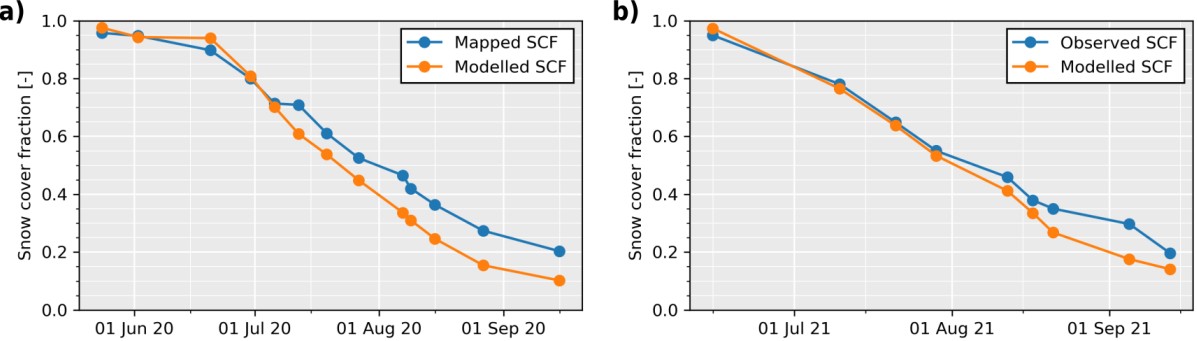

**Figure E6.** Modelled and mapped snow cover fraction (SCF) for **(a)** 2020 and **(b)** 2021 based on mapped snow cover from planet satellite imagery and as modelled by the mass balance model.



*Code and data availability.* All isotopes data are available under (Müller, 2023b), weather data under (Müller, 2022) and river data under (Müller and Miesen, 2022). The codes for the glacio-hydrological model were written in python using Jupyter Notebook and are provided in the Supplement as well as all PEST calibration files.

*Author contributions.* T.M. conducted all the data collection and data analysis, produced all the figures and wrote the manuscript draft. B.S. 830 proposed the general research topic and acquired the funding. S.L. and his team organised field work logistics. M.F. organized all mass balance related field work and provided point and glacier-wide surface mass balance data for the Otemma glacier. B.S. and S.L. jointly supervised the research. M.F., S.L. and B.S. edited the manuscript draft version. All authors have read and agreed to the current version of the manuscript.

*Competing interests.* The contact author has declared that none of the authors has any competing interests.

*Acknowledgements.* T.M. and M.F. thank Vera Girod who helped with the field work in 2020 and Valentin Tanniger who carried out the dye tracing work. T.M. also thanks all Bachelor, Master and PhD students from the University of Lausanne who helped with data collection, and in particular Floreana Miesen who organised field logistics and participated in field data collection on the Otemma glacier.

*Financial support.* This research has been supported by the Schweizerischer Nationalfonds zur Förderung der Wissenschaftlichen Forschung (grant no. 200021_182065).



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
