# Peer review of "Separating snow and ice melt using water stable isotopes and glacio-hydrological modelling: towards improving the application of isotope analyses in highly glacierized catchments"

_EGUsphere, 2024_

## Author Response (AR1)

**Comment on Egusphere-2024-631 - Anonymous Referee #1**

We thank reviewer 1 for the detailed comments. Below we quote the original comment followed by our response. The bold font has been introduced by us for readability.

General comments

**Comment**

*This work presents an interesting modeling effort in a Swiss glacierized catchment, which involved a water stable isotope module. Although the authors provided detailed information regarding the data, model structure, and sensitivity analysis, the novelty of this work needs to be further emphasized. Specifically, the authors should explain* **how the assessments in this study can inform** hydrological modeling **in glacierized basins.**

*In the introduction, the authors listed a couple of* **modeling challenges for glacierized basins***, which is good. My major suggestion for the authors is to improve their results by providing* **more evidence on how** *the built isotope-glacio hydrological model or the* **assessments conducted in this study can help to address such modeling challenge***s.*

*Currently, the paper reads more like a description of the model and data. How the model and their findings will contribute to the research community is not well evaluated, which falls below the standard of an original research paper. Additionally, the authors concluded that the isotope-glacio hydrological model* **didn't guarantee better separation results of the water shares** *and showed similar performance to a normal glacio-hydrological model. Even though the authors tried to emphasize that combining isotopes with glacio-hydrological modeling* **enhanced hydrologic parameter identifiability,** *this aspect was not adequately assessed in the results. Given the added model parameters and model configuration complexity, why an isotope-glacio hydrological model is evaluated in this study?*

*What are the benefits of integrating isotopes into a well-verified glacio-hydrological model in the study area? Perhaps, the authors might consider adding a benchmark model for improved assessments on the influences (or benefits) of combining isotopes.*

**Answer**

We thank the reviewer for their careful read of our manuscript and detailed comments. We understand that the reviewer's main suggestion is to emphasize the usefulness of the isotopic module to inform a glacio-hydrological model and to discuss how the findings of this work contribute to address some of the major modelling challenges for glacierized catchments. Below, we first answer these concerns before giving a list of detailed changes to address them.

*Lack of emphasis of how the work benefits glacio-hydrological modelling:* We agree that the paper is quite complex and addresses multiple topics, which may be difficult to follow for the reader. Although we developed a complex model, the focus of the paper is not directly on assessing the benefits of informing a glacio-hydrological model with isotopes, but rather on illustrating the complexity and evolution of the isotopic composition of different end-members in a glaciated catchment. **Our main finding** is that using isotopes to separate the shares of snow and ice in a glaciated catchment in the Alps is not advisable and

that a "simple" mass-balance model may provide better estimates (see Fig. 10 of the original and revised manuscript). This is clearly stated in the revised conclusion.

*Benefits of a combined isotopic-glacio-hydrologic model*: As we developed a complex model, we also address the benefits of the combined isotopic and glacio-hydrological model, but the improvement in terms of hourly discharge estimations is clearly limited. This statement is indeed not clearly illustrated in the paper and we provide more information in the revised manuscript. In particular, we show how the hydrological transfer module and hence the modelled discharge and the model parameters are affected if the model is calibrated against discharge data alone, or with discharge and isotopes data simultaneously. Nevertheless, as already shown in Figure 11 (of the original and revised manuscript), we show that isotopes can inform glacio-hydrological models, not so much in term of discharge volumes, but in terms of the model structure and of how the model reacts to rain events. This may be useful to improve glacio-hydrological models focusing on the response to (extreme) rain events or how rain-on-snow affect the hydrological response for instance.

Finally, we emphasize the fact that the isotopic module may not benefit a well-informed glacio-hydrological model for discharge simulation, because snow and ice compositions appear to have similar values during a large part of the melt season. For this reason, our hydrological and isotopic model may provide more useful information for snow-dominated catchments where ice is not present, as already proposed by Ala-Aho. For this reason, we detailed in the discussion the sensitivity of the isotopic module to different parameters. This discussion may be useful for readers focusing more on snow isotope modelling and also provides complementary analysis to the model proposed by Ala-Aho.

In short, the paper addresses 3 main questions (which are now mentioned in the introduction, answered in the conclusion and that we address in the discussion in separate sections) :

- Can we model the catchment-integrated snow and ice melt isotopic compositions during a whole year using a parsimonious model ?
- Can we separate snow and ice shares in streamflow at catchment-scale based on water isotopes data alone ?
- What are the benefits of integrating isotopes in a glacio-hydrological models and can it contribute to better separate modelled snow and ice melt shares ?

**Main changes :**

We revised the manuscript and attempted to address all the reviewer's comments. In particular, the following changes were made :

- Introduction : We adapted the discussion and provided a more focused introduction to the paper in the last few paragraphs. We also moved the part which discussed isotopic theory to the method (see new Sect. 2.5).
- Methods : We lightly adapted the method section. We encourage the reader to skip most parts that are not especially innovative while directing the reader to the isotopic model part or the calibration. We explained more clearly the stepwise calibration. We provided a bit more explanation on our assumptions about ROS isotopic mixing.
- Results: we simplified some parts of the results, which were moved to the discussion or removed. Section 4.3 (Hydrological transfer module results) and Figure 6 were modified to discuss modelled

stream discharge when the model is calibrated with or without isotopes. Some more literature for comparison is also provided.

- Discussion : we re-organized the sections in the discussion.
    - Section 5.1 (Mass balance model limitations) : we provided a longer discussion on snow redistribution. We reduced the discussion on minor processes not included in the model
    - Groundwater section was moved earlier, from Sect. 5.4 to Section 5.2.
    - Section 5.3 (Isotopic model sensitivity analysis) : we adapted some parts, especially regarding ROS. We moved the discussion about d-excess in section 5.3.4. We moved, simplified and adapted the previous section 5.5 (Towards a better estimation of the snow isotopic composition) to a subsection 5.3.6 (Snow isotopic module calibration challenges and opportunities).
    - Section 5.4 (Water isotopes and end-member mixing models in glacierized catchment) was renamed and moved here from previous Section 5.6 (Mixing model limitations). We moved also previous Figure 11, which is not Figure 10. We simplified the figure and removed the part showing results without transfer. The message was too complex and missed a bit the main focus of the paper. We also modified the text to focus more directly on the use of isotopes to separate snow and ice.
    - Section 5.5 (Water isotopes for hydrological routing in glacierized catchment) was moved and adapted from previous Section 5.3 (Limitations of the hydrological transfer model). We adapted the text to focus more on the use of isotopes to better inform models for the parametrization of the routing scheme. Figure 10 was moved here and renamed Figure 11.
    - The three last sections now address the three main questions of the introduction.
- Conclusion: we simplified the conclusion and removed the long final part, which is now summarized in 3 main bullet points answering the questions from the introduction.

Detailed comments

**Comment 1**

*As listed in lines 82-85, some processes were added in the model. However, the impacts of the additional processes on the model simulations were not well assessed. I would suggest adding a reference model in which these processes were not involved for the assessment.*

**Answer 1**

Indeed, this was not clearly shown in the paper.

**Changes** : we removed the lines describing the isotope module from the introduction, which we found too detailed and is now found in the method. As suggested by the reviewer, we adapted Figure 6 to include the modelled discharge when including isotopes for calibration or not. We also adapted Section 5.3 which now discusses the benefits of including isotopes more precisely.

**Comment 2**

*Reference literature for many equations was missing. For example, Eqs. 3, 4, 5-7, 9, and 23.*

**Answer 2**

Equations 3 to 9 were developed for the model needs and are not referenced from previous literature. As they may appear difficult to "read", we illustrated their behavior in Appendix (Figure D1).

**Changes** : We directed the reader to the Appendix for a more visual understanding of the results of those equations.

**Comment 3**

*Eq. 28, please add calculations for $\phi Rain$ and $\phi Ros$.*

**Answer 3**

**Changes**: added as requested.

**Comment 4**

*The model calibration procedure should be better described. Is this a stepwise calibration or a simultaneously multiple-objective calibration? How were the competitions between the performance of discharge simulation, isotope simulation, and snow simulation dealt with in the calibration?*

**Answer 4**

The model calibration is stepwise, each module is calibrated separately using multiple objective functions. This is detailed in sections 3.1.7, 3.2.4, 3.3.6. Our goal was not to develop a fully simultaneously calibrated model but rather to assess the added benefits of the hydrological transfer and isotope module when estimating discharge volumes and snow and ice-melt fraction. For this reason, the calibration was performed separately for each module, and there is, therefore, no competition or weights between the calibration of the 3 modules.

We understand this approach may appear unconventional but we believe it better illustrates the benefits of each module.

**Changes** : We made this clear at the beginning of section 3. We also specified this in the introduction (last two paragraphs).

**Comment 5**

*In Fig. 4, the coefficients of determination of the linear curves appear to be very low. Do the lines make any sense for the modeling? Was any trend-test conducted?*

**Answer 5**

Indeed, the lines are only here to show that no trend with elevation can be found here. We believe they help the reader to better visualize the results. These lines are not used in the modeling.

**Changes** : no change was provided.

**Comment 6**

*In Fig. 6, sub-daily fluctuations of daily discharge can be seen from the plots. Are these daily data (one point per day)?*

**Answer 6**

No, the data are hourly. This is stated earlier in the manuscript.

**Changes** : we specified "hourly measured discharge" in the legend.

**Comment 7**

*In Section 5, please add discussions on the model's performance in simulating snow, ice, isotopes, and discharge, with respect to comparisons with peer research.*

**Answer 7**

We agree that we could provide a comparison for discharge. However, the goal of the paper is not to create yet another glacio-hydrological model, but the model is rather used as an example to illustrate snow and ice-melt processes and to assess the use of isotopes in glaciated catchments. Also, in our case, the enhanced temperature index (ETI) mass balance model was calibrated without discharge data. Discharge data were only used for the routing scheme. This is a particularity of the model, as most ETI models are calibrated against discharge, which clearly improves their ability to simulate discharge. Those models have a Nash-Sutcliff efficiency (NSE) ranging form 0.7 to 0.9 depending on the model complexity (e.g., Schaefli et al., 2005; Huss et al., 2008; Magnusson et al., 2011).

Some energy-based models with complex snow processes may be calibrated without discharge (for the mass balance) and have NSE in the order of 0.6 to 0.9. We only found one paper by Shakoor et al. (2018) that uses Alpine3D, a 3D distributed model for mountain hydrology, which simulates snow processes in details and uses an adapted temperature-index scheme for snow melt without discharge calibration. Here they obtained a NSE of 0.77 but their model was significantly more complex.

**Changes** : we adapted Section 4.3 to provide the literature according to the above comments.

**Comment 8**

*In Fig. 10, what are the reasons for the much higher isotopic composition of measured discharge than the pre-event? Is it due to evapotranspiration or contributions from other sources? From the plots, a very low fraction of pre-event can be observed in the discharge. If the fraction of pre-event is low, is this plot good evidence to illustrate the response of discharge to rain events?*

**Answer 8**

The isotopic composition of measured discharge increases during rain events due to rain, which has a much higher isotopic composition than the pre-event discharge (composed mainly of snow and ice-melt). During rain events, the isotopic composition of the discharge increases due to the rain mixing with snow and ice-melt. This plot shows that, especially during the early- to mid-melt season (July), the discharge is affected by the rain water during more than a week, which appears in the isotopic signal but is hardly visible in the discharge volumes.

**Changes** : we indicated the isotopic composition of rain during those events in Figure 10 and improved the text of the legend. We  revised the discussion to make our point clearer.

**Comment 9**

*In lines 703-709, as commented before, what are the benefits of involving isotopes for hydrological modeling in this study area?*

**Answer 9**

**Changes** : Here, as discussed in the answer to the general comment, we adapted Figure 6 to compare the model results with and without involving isotopes in the calibration. We also better separated the findings of this study in the discussion with a section dedicated to the use of isotopes to inform models, especially for water routing.

**Comment 10**

*In Section 5.4, why not give an estimate for the fractional contribution of groundwater to discharge? Without a contribution number, it is a very weak discussion on the role of groundwater.*

**Answer 10**

**Changes** : we provided a fractional estimate for groundwater

**Comment 11**

*Sensitivity of model parameters to the assumptions should be added. Also, please add a sensitivity analysis of model parameters to the model configuration of isotopes, as well as the influences of added processes on model parameter identifiability.*

**Answer 11**

Since the goal of the paper is not directly to propose a "new" glacio-hydrological model, we do not want to further extend the manuscript by providing a full detailed sensitivity analysis of all parameters. We restrain the sensitivity analysis to the evolution of the snow isotopic composition, which is one of the novelties of the paper and may be useful for further development of isotopic models. In addition, the use of isotopes to restrain model parameters would be interesting only if the model was calibrated in a simultaneous multiple-objective calibration using all data, which is not the case. An interesting paper by He et al. (2019), already provides such an approach, which is not our aim here.

**Changes** : With the new structure of the manuscript and more focused discussion we hope that the paper provides enough insight on how isotopes may or may not be useful for glacierized catchment.

**Comment 12**

*Perhaps the conclusions can be further improved by emphasizing their novel findings better. "(iii) assess how water isotopes can be used to estimate the proportions of different water sources at the outlet." This might need to be better interpreted.*

**Answer 12**

We revised the conclusion, reflecting also the improved paper structure, highlighting the main outcomes as discussed previously.

**Changes :** We adapted the conclusion, which now better focuses on the 3 questions introduced in the introduction, which correspond to the 3 last sections of the discussion.

**Comment 13**

*In Table A1, why are there multiple temperature lapse rates?*

**Answer 13**

The temperature lapse rate function is estimated using a "Gaussian"-like shape, which needs 4 parameters. **Changes** : we adapted the name of the parameters.

**Comment on Egusphere-2024-631 - Anonymous Referee #2**

We thank reviewer 2 for the detailed comments. Below we quote the original comment followed by our response. The bold font has been introduced by us for readability.

General comments

**Comment**

*The work presents glacio-hydrological model incorporating stable water isotope data in the simulations. The work is done at a glacierized catchment in the Swiss Apls, and uses a multitude of data sources for model calibration, which is performed in different steps. They found out that the time-variable snowmelt isotope values converge with ice melt isotope values late in the season, bringing about the typical problem in isotope-based mixing model analysis in glacial catchments. However, the tracer-aided model can provide additional insight to the runoff generation processes and snow vs glacial melt contributions.*

*I find that the work is a interesting and ambitious case study leveraging of multiple data sources for model calibration. I liked the fresh approach of using transfer module approach, circumventing the need for explicit conceptual storages for soil and GW most often used in hydrological modeling. Though also that approach resulted in distribution parameters to calibrate,* **and the stepwise calibration approach was a bit difficult to follow.** *Because of the explicit model description and multitude of data sources, I found the paper fairly long, and suggest below* **some parts that could be considered leaving out.** *I recommend the authors address my comments below publishing the work.*

**Answer**

We thank the reviewer for their detailed comments, in particular regarding the isotope modelling framework, which nicely complements the comments from the first reviewer. The paper is indeed relatively long, and the approach may appear rather unconventional as our aim was not directly to propose (yet) another model, but rather to use a detailed model to illustrate how water isotopes evolve in a glaciated catchment and to discuss where they may be more useful.

In short, the paper addresses 3 main questions (which are now mentioned in the introduction, answered in the conclusion and that we address in the discussion in separate sections) :

- Can we model the catchment-integrated snow and ice melt isotopic compositions during an entire year using a parsimonious model?

- Can we separate snow and ice shares in streamflow at catchment-scale based on water isotopes data alone ?

- What are the benefits of integrating isotopes in a glacio-hydrological models and can it contribute to better separate modelled snow and ice melt shares ?

**Main changes :**

We revised the manuscript and attempted to address all the reviewers comments. In particular, the following changes were made :

- Introduction : We adapted the discussion and provided a more focused introduction to the paper in the last few paragraphs. We also moved the part that discussed isotopic theory to the method (see new Sect. 2.5).
- Methods : We lightly adapted the method section. We encourage the reader to skip most parts that are not especially innovative while directing the reader to the isotopic model part or the calibration. We explained more clearly the stepwise calibration. We provided a bit more explanation on our assumptions about ROS isotopic mixing.
- Results: we simplified some parts of the results, which were moved to the discussion or removed. Section 4.3 (Hydrological transfer module results) and Figure 6 were modified to discuss modelled stream discharge when the model is calibrated with or without isotopes. Some more literature for comparison is also provided.
- Discussion : we re-organized the sections in the discussion.
  - Section 5.1 (Mass balance model limitations) : we provided a longer discussion on snow redistribution. We reduced the discussion on minor processes not included in the model
  - Groundwater section was moved earlier, from Sect. 5.4 to Section 5.2.
  - Section 5.3 (Isotopic model sensitivity analysis) : we adapted some parts, especially regarding ROS. We moved discussion about d-excess to section 5.3.4. We moved, simplified and adapted the previous section 5.5 (Towards a better estimation of the snow isotopic composition) to a subsection 5.3.6 (Snow isotopic module calibration challenges and opportunities).
  - Section 5.4 (Water isotopes and end-member mixing models in glacierized catchment) was renamed and moved here from the previous Section 5.6 (Mixing model limitations). We also moved Figure 11, which is not Figure 10. We simplified the figure and removed the part showing results without transfer. The message was too complex and missed a bit the main focus of the paper. We also modified the text to focus more directly on the use of isotope to separate snow and ice.
  - Section 5.5 (Water isotopes for hydrological routing in glacierized catchment) was moved and adapted from previous Section 5.3 (Limitations of the hydrological transfer model). We adapted the text to focus more on the use of isotopes to better inform models for the parametrization of the routing scheme. Figure 10 was moved here and renamed figure 11.
  - The three last sections now address the three main questions of the introduction.
- Conclusion: we simplified the conclusion and removed the long final part which is now summarized in 3 main bullet points answering the question in the introduction.

Detailed comments

**Comment 1**

*L135: specify how measured? average snow depth reduction in the ablations stakes? Where the density 900 assumption comes from?*

**Answer 1**

We mean here that we measured at each ablation stake the decrease of the ice/snow surface in centimeters, 3 and 7 times during the seasons 2020 and 2021. The data at each stake are then used for the model calibration without averaging. The ice density of 900 comes from common literature.

**Changes** : modified the sentence. Provide literature for the ice density.

**Comment 2**

*L161: Shaded location may not be enough, if average is 7, highs can get much more than that. Did you examine the potential evaporation effects? No paraffine oil of other fine tubes to prevent evaporation?*

**Answer 2**

No parafine was used here. Our results also showed no deviation of the samples from the local meteoric water line which indicates limited evaporative losses. In addition, von Freyberg et al. (2020) showed a deuterium difference of about 1‰ for a two to three weeks period at such temperatures. We therefore believe that no significant evaporation should have occurred. Our results also showed no deviation of the samples from the local meteoric water line, which should indicate limited evaporative losses.

**Changes:** no changes were made here to avoid increasing the manuscript length.

**Comment 3**

*L209: where is this Fig. D1d?*

**Answer 3**

Not sure about this question. It is at the end of the paper in the Appendix.

**Changes:** no changes were made here.

**Comment 4**

*L221: is the snow redistribution scheme based on some existing work? I find no refs here. Wind speed is not a factor?*

**Answer 4**

No, we made our own function, which is a very rough but simple approach that which conserves mass over the catchment.

**Changes**: we added a short sentence in the discussion (section 5.1) and moved a sentence from the results there too.

**Comment 5**

*L278: how were the other two field datasets used in the PEST inversion?*

**Answer 5**

The three datasets were calibrated simultaneously using a multi-objective function.

**Changes**: we added two sentences in Sect. 3.1.7 to clarify this.

**Comment 6**

*L302: how good is the regression goodness? Typically, you would miss the extreme values, the lows in particular*

**Answer 6**

This is provided in the results section 4.4 and Figure 7. Indeed, the low values have a relatively strong influence on the regression, which was also shown in the sensitivity analysis in Figure 9 (red area). The coefficient of determination is 0.85.

**Changes** : we did not change this as this is already mentioned in the result and discussion.

**Comment 7**

*L328: I'm not convinced by the assumption. Why would a think snowpack be less ripe by assumption? It can be equally isothermal and retain liquid water as a smaller pack. Do you have any data to back up this assumption? See my later comment on this*

**Answer 7**

We mostly answered this essential comment in comments 17 and 18 below.

Here we would like to stress that we are talking about the mixing of the isotopic signal and not directly of the water retention of the snow. This assumption was based on the work of Juras et al (2019), who showed that rain water flowing through a thicker, colder, snowpack was isotopically less affected by the snowpack isotopic composition than a shallower, ripe, snowpack (meaning there is less mass exchange between rain and snow in this snowpack).

There is indeed no clear evidence that a thicker snowpack is less ripe by assumption. We can only say that a 2000 mm snowpack only occurs at high elevation where temperature and melt remain very low, so that ripe conditions are unlikely. Finally, the physical basis of this function is indeed disputable, but the calibration of this function led to satisfying results, and this formulation was therefore retained. It has the advantage of being simple; estimating the hourly snow energy state at each grid cell is numerically too complex.

We discuss this essential topic in subsection 5.2.3.

**Changes** : we adapted the text of Section 3.2.3 for more clarity and to avoid confusion regarding isotopic mixing. We provided the reference to the work of Juras et al (2019) regarding the isotopic mixing of rain in ripe and unripe snowpacks. We also adapted the discussion in sect. 5.2.3. (see answers to comments 17 and 18).

**Comment 8**

*L333-338 :The rules seem a bit subjective. Why did you not compare your simulated i_sp with the depth (or SWE even better) averaged snow pit d2H? Or use the snowmelt samples you had? Instead of this more "soft" calibration.*

**Answer 8**

We unfortunately did not have enough data for a proper calibration against the snow pit or snowmelt measurements. This would have been indeed more straightforward but those data are hard to acquire, and we lacked data during the whole summer. We nonetheless verified that our calibration fitted with the average value of the snow-pits. Indeed, the modelled snowpack δ2H at the snowpits were around -119‰ for both pits compared to two mean measured snowpits δ2H of -125‰ and -117‰.

In addition, this "soft" approach also provides some advantages that may be further investigated in future studies. It does not require snowpit samples, which are difficult to acquire and may vary significantly due to local snow conditions. By using stream samples, we can much more easily obtain a catchment-averaged value of snowmelt, and we can acquire data during the whole season and especially during the end of the melt season when snow samples are almost impossible to acquire at high elevations. We will add this comment to the results. The lack of data for calibration is already discussed in the discussion, where we also encourage to validate this approach with depth-averaged snowpit data.

**Changes** : We adapted this section in the new manuscript (see also answer 9). We provided the value of the snowpits mentioned above in the results (Sect. 4.5). We also changed the discussion by moving previous Section 5.5 to Section 5.2.6, which now focuses more on the snow isotopic module calibration and is clearer than previously.

**Comment 9**

*L338 – L 345: this is very difficult to understand without seeing the timeseries data*

**Answer 9**

Yes, this part was difficult to follow. The calibration procedure was also not clearly described. Indeed, we also compare modelled and measured stream isotopic compositions to calibrate the isotope module. This is now better explained.

**Changes**: We removed this comment, which was not very relevant for the paper. We also added a new explanation that was lacking in the paper before (see also answer 8).

**Comment 10**

*L365: not sure I understand this, not familiar with the kinematic subsurface saturated flow concept: do you mean that all water delivery from the hillslope is darcian-type GW flow, with instantaneous recharge. No overland flow?*

**Answer 10**

Yes, we assume that all water infiltrates in the sediments, which are very coarse. However, the groundwater flow in those sediments is fast due to the slope and high hydraulic conductivity. Even if no overland flow occurs, water is rapidly conveyed downslope. No overland flow can be observed after rain events, only tributaries coming from lateral glaciers can be observed. This was observed and discussed in previous research (Müller et al., 2022).

**Changes** : we added a sentence here.

**Comment 11**

*L447: do you propose that snow surface samples are representative of the snowfall precipitation over winter? That would be the logic behind the Beria et al statements, where the isotope variability in snowfall events over winter is higher than variability in snowmelt.*

**Answer 11**

At this stage in the result, we only comment the results. But snow surface samples do represent at least one "layer" of snow which accumulated during the winter. It is, however, not homogenous, as snow melts faster in the lower part of the catchment. Also due to snow sublimation and vapor deposition, vapor exchanges within the snowpack occurs. As a result, surface snow samples may have deviated from the composition of the snow event. Yes, surface snow samples represent more or less the variability of the composition of snow events in winter, which then homogenizes due to vapor exchanges and during melt. This is, to our understanding, also what is discussed in Beria et al.

**Changes** : we provided a brief sentence about surface snow samples in the new section 5.2.6

**Comment 12**

*L452: Interesting data, sublimation can be causing this. There is pretty high variability in your snow surface d-excess values, considering your sampling strategy. If the surface snow is deposited from the same storm, and experiences the same atmospheric conditions after deposition, the d-excess values should be pretty similar. I recommend to discuss what pre and post depositional processes might cause this.*

**Answer 12**

Indeed, d-excess appears variable, and no trend can be observed. This is probably because the snow at different elevations represents different snowfall events, and the rate of sublimation/deposition may vary spatially. We briefly tackle this process in the discussion (L754) and also direct the reader to more focused research on the topic (e.g. Sprenger et al. 2024 https://doi.org/10.5194/hess-28-1711-2024).

**Changes** : we discuss this shortly in section 5.2.4.

**Comment 13**

*L453: to me it looks like there is a positive correlation between H2 and elevation, contrary to what one would expect.*

**Answer 13**

As the coefficient of correlation is very low ($R^2$ = 0.26) for dH2 of snow surface and elevation, we find it difficult to confirm a clear trend, also for the reasons discussed above. It is also contrary to what one would expect, as precipitation dH2 typically decreases with elevation (see Beria et al). One explanation could be that at high elevations, surface snow represents a snowfall event from the end of the winter period, which typically has a higher temperature and thus higher dH2 value, while, at lower elevations, the end of winter snowfall has already melted and the surface snow actually represents snowfall from the mid-winter season, which has a typically lower dH2 value. But this cannot be confirmed with our data.

**Changes** : we added the value of R2.

**Comment 14**

*L473: why did you not calibrate one parameter set for both years? This would be the typical approach where parameters are assumed invariant in time.*

**Answer 14**

Since both winters and summers showed different weather conditions (one drier and one more humid summer for instance), model parameters may vary as a simple temperature-index model cannot account for such conditions. In addition, here, the goal was to obtain a precise estimation of the isotopic signal for each year of the study, and not to calibrate the model for projection in the future. As such, a yearly calibration leads to better results.

**Changes** : we added a sentence similar to the answer

**Comment 15**

*L484: can you justify how SWE simulations were deemed good? RMSE of ~100 mm w.e. seems pretty high? What was your catchment average SWE, so you could put the error to perspective*

**Answer 15**

The total snow melt of each year was 1860 and 1527 mm, this is indicated in the text one paragraph below. Although a RMSE of 100 mm w.e. seems high, note the mean error is less than 10 mm, and also the total cumulated melt is comparable to measured discharge, which indicates that our model performs well at the catchment-scale. Also, those results are similar to other advanced snow models where RMSE values are in the range of 75 to 150 mm w.e. for such elevated catchments (Mott et al., 2023). The RMSE error seems to be mainly due to the local conditions at each snow depth measurement point. Also, our model remains simple, with no spatially resolved snow redistribution process, which probably results in a smoother SWE accumulation than in reality. Also note that our model also needs to correctly simulate snow on the steep hillslope which, as a tradeoff, may also increase the model error on the glacier where snow depth data were obtained.

**Changes** : We added a reference to the work of Mott et al.(2023) for comparison.

**Comment 16**

*Fig 8a: add in the legend "snow melt composition" not to confuse with snowpack*

**Answer 16**

**Changes** : changed as proposed

**Comment 17**

*Fig 8: what is the physical process that would explain the snowmelt isotope values to become gradually more depleted at times your 2021 simulations?*

**Answer 17**

This comment also relates to your previous comment #7 about our function to mix ROS with the snowpack.

We believe snowpack and snowmelt increase in summer is due to rain-on-snow (ROS), which partially integrates/mixes with the snow. Juras et al (2017), in particular, also showed, using an artificial setup, that snowpack samples showed a considerable increase in dH2 after a rain (here sprinkling event). This suggests that mass exchange between the snow and rain occurs, therefore increasing the dH2 composition of the snowpack. This observation is an essential and highly difficult process to simulate with a simple model (without simulating the snow internal temperature for instance). For this reason, we dedicated a

subsection (5.2.3) to this process. We agree that the model may need further development, but we at least showed and discussed the importance of ROS to simulate snow isotopic composition. Finally, we want to stress that the increase in snowmelt isotopic composition during the summer may be due to both ROS and snow fractionation. In our results, it seems that ROS influences snow composition more than fractionation, especially when the snowpack is shallow. This is shown in the sensitivity analysis in Figure 9.

In 2021 in particular snowmelt d2H increases shortly during precipitation event. This is due to the short-term deposition and subsequent melt of summer snow at low elevations (where snow was absent), which has a higher d2H composition than the older remaining winter snowpack at higher elevations. This fresh snow disappears in a few days after which the snow melt d2H gradually returns to the composition of older snow.

**Changes** : The behavior of the snowmelt d2H over the season is shown in detail in the sensitivity analysis. We added a sentence in the result to comment the increase of the snowmelt d2H during the melting season. We also added the last comment above in the results at the end of section 4.5.

**Comment 18**

*L534: this seems counter intuitive, that a bigger snowpack would mix less with rain, ie. deliver more rain water throught the snowpack, that a smaller pack. I think liquid water retained in the snowpack would be important to consider here. Snowpack with 2000mm w.e. is massive, and conceptually difficult to see that storms of your magnitude (mostly 10 mm/day) would seep through the snowpack. One modeling Rule of thumb for snow water retention is 5% of w.e., which in your snowpack would be 100mm of water. Is there any other explanation why you get less enrichment in thick snowpacks?*

**Answer 18**

Please also read our answers to comment 7 and 18.

Yes, this is definitely a weakness of the model which we borrowed from Ala-Aho. Here, they assume that the isotopic signal of the rain is completely integrated in the snowpack (and the rain water is then released with a composition similar to snow). We tried to improve this simplification in our model with a function that only partially mixes the isotopic signal of the rain with the snow.

We want to make some aspects clearer. Here we talk only about how the rain isotopic composition mixes within the snowpack through mass exchange processes. In our model, snow water retention is not clearly modelled and all rain passing through the snowpack is released depending on a transfer function, which is a function of snow depth. So, in a thicker snowpack, the rate of water release will be delayed and smoother than in a thinner snowpack. But in both cases the release of water is equivalent to the rain input.

However, while traveling through the snowpack, the isotopic composition of the rain water will partially mix with the snowpack. Juras et al. (2017) showed that in a cold (-1°C) but thicker snowpack (50 cm), the rain isotopic composition seeping through the snowpack retains its original composition much more than in a shallower isothermal snowpack. The rate of infiltration was also very fast (10 cm/min or 6m/hour) and the release almost not delayed. They attribute this effect to the formation of preferential flow in the non-ripe, cold, snowpack, leading to less isotopic mixing.

Our function attempted to mimic this effect by simply allowing the rain water through the snowpack to only partially isotopically mix. This is done in a very simple way using SWE as an indicator of snow ripeness.

This function is debatable indeed, but SWE was already available, while estimating the snow energy state seemed too complex for such a model. We attempted to discuss this essential topic in subsection 5.2.3.

**Changes** : the sentence in the results was removed because it had a discussion character. It was moved to the discussion in section 5.2.3. We also adapted the discussion according to the answer above.

**Comment 19**

*L565-575: good discussion about the uncertainties in sublimation.*

**Answer 19**

Thank you !

**Comment 20**

*L588: including the objective functions, and in particular how the snow extent maps were numerically compared with simulations, would be an important addition to the methodology.*

**Answer 20**

This is described in section 3.1.7 Calibration. We do not want to go into more detail as the description seems to be clear, since figures and codes are available.

**Changes** : we directed the reader to Figure E1 to E4. We slightly adapted the text in the methodology.

**Comment 21**

*Chapter 5.3 This is interesting discussion, but for the benefit of shortening the paper, could be left out without compromising the main findings in the paper.*

**Answer 21**

Ok, thank you for your comment. The paper tackled essentially two topics: to illustrate snow and icemelt isotopes processes in a glaciated catchment on one hand; and how isotopes can inform a glacio-hydrological model. This discussion mainly informed the latter and was requested by another reviewer.

**Changes** : we adapted section 5.3 (now section 5.5) to focus more on how isotopes may inform glacio-hydrological model

**Comment 22**

*L695-702: the degrees of freedom caused by the model complexity are apparent in this discussion. The stream isotope response can be explained by many independent processes. This is not a critical comment as such, but you system is very complex to conceptualize even with isotope data and models.*

**Answer 22**

Yes, indeed, and this is why we provided Figure 9 in particular.

**Changes** : We provided in Figure 6 a comparison of model results with or without isotope calibration. We also mention in section 4.3 problems of model equifinality. We also rearranged the discussion for more clarity.

**Comment 23**

*L704: in the objective function?*

**Answer 23**

Yes

**Changes** : we adapted section 5.3 (now section 5.5) to focus more on how isotopes may inform glacio-hydrological model. We provided in Figure 6 a comparison of model results with or without isotope calibration in the objective function.

**Comment 24**

*L751: what do you mean by outlet snowmelt d2H? The stream value, of d2H leaving the snowpack (the meltwater)?*

**Answer 24**

Yes, we meant the catchment-integrated snow melt d2H at the outlet.

**Changes**: we adapted the term. Note that this section was moved to Sect 5.2.6

**Comment 25**

*L791: also the model would be good to validate with either snowpack or snowmelt samples.*

**Answer 25**

Yes, we adapted the discussion (see also answer 8).

**Comment 26**

*L801: not clear why you did not collect the bulk snowpack samples in this case: they are easier to get as a by product of SWE measurements than a snow pit profile.*

**Answer 26**

In our approach, we needed the isotopic composition of precipitation to estimate the snowpack isotopic composition spatially and during the whole season. I am not sure we could obtain the same seasonal modelling of snow isotopes based on snowpack data, which become hard to acquire during the late season. The model may be calibrated with snowpack samples alone, but this approach may be as much (or more) time consuming, and we did not acquire many snowpit data unfortunately (see also Answer 8). Ideally, using both precipitation samples and snowpack data is clearly advisable.

**Changes** : we adapted the conclusion and mention the limited snowpack data available.

**Comment on Egusphere-2024-631 - Anonymous Referee #3**

We thank reviewer 3 for the detailed comments. Below we quote the original comment followed by our response. The bold font has been introduced by us for readability.

General comments

**Comment**

*This manuscript reports an interesting combination of a glacio-hydrological and isotopic model to estimate the seasonal shares of snowmelt and ice melt in the stream waters of a high-elevation catchment in the Swiss Alps. The authors well described their dataset, the model with the various modules and the sensitivity analysis, and provided some useful recommendations for future studies including the collection of samples for isotopic analysis from different water sources in glacierized catchments. Overall, I think this is a valuable research paper that deserves to be published. However, some modifications are needed (please see the specific comments) before the acceptance of this paper. Among these changes, in agreement with reviewer 1, **I think the authors should better emphasize the advantages of integrating isotopic observations and an isotopic model into the glacio-hydrological model that was used.** Indeed, some results and the discussion highlight more the challenges of using the isotopic tracers than the usefulness of their integration into the glacio-hydrological model.*

*Furthermore, since the manuscript is quite long, **I recommend to the authors to better emphasize the novelty of the manuscript** (compared to other research papers) **in the abstract and the conclusions,** to **shorten some paragraphs in the conclusions** and better **highlight the key findings.**￼*

**Answer**

We thank the reviewer for their careful reading of our manuscript and the detailed comments, and especially their suggestion to better structure and shorten the manuscript. As also answered to the previous reviewers, the focus of the paper was not directly to assess the benefits of informing a glacio-hydrological model with isotopes, but rather to illustrate the complexity and evolution of the isotopic composition of different end-members in a glaciated catchment.

As suggested by the reviewer, we emphasized this in the introduction, discussion and conclusion to better highlight the key inputs of the paper. In short, the paper addresses 3 main questions (which are now mentioned in the introduction, answered in the conclusion and that we address in the discussion in separate sections) :

- Can we model the catchment-integrated snow and ice melt isotopic compositions during an entire year using a parsimonious model ?

- Can we separate snow and ice shares in streamflow at catchment-scale based on water isotopes data alone ?

- What are the benefits of integrating isotopes in a glacio-hydrological models and can it contribute to better separate modelled snow and ice melt shares?

**Main changes :**

We revised the manuscript and attempted to address all the reviewer's comments. In particular, the following changes were made :

- Introduction : We adapted the discussion and provided a more focused introduction to the paper in the last few paragraphs. We also moved the part that discussed isotopic theory to the method (see new Sect. 2.5).
- Methods : We lightly adapted the method section. We encourage the reader to skip most parts that are not especially innovative while directing the reader to the isotopic model part or the calibration. We explained more clearly the stepwise calibration. We provided a bit more explanation of our assumptions about ROS isotopic mixing.
- Results: we simplified some parts of the results, which were moved to the discussion or removed. Section 4.3 (Hydrological transfer module results) and Figure 6 were modified to discuss modelled stream discharge when the model is calibrated with or without isotopes. Some more literature for comparison is also provided.
- Discussion : we re-organized the sections in the discussion.
  - Section 5.1 (Mass balance model limitations) : we provided a longer discussion on snow redistribution. We reduced the discussion on minor processes not included in the model
  - Groundwater section was moved earlier, from Sect. 5.4 to Section 5.2.
  - Section 5.3 (Isotopic model sensitivity analysis) : we adapted some parts, especially regarding ROS. We moved the discussion about d-excess in section 5.3.4. We moved, simplified and adapted the previous section 5.5 (Towards a better estimation of the snow isotopic composition) to a subsection 5.3.6 (Snow isotopic module calibration challenges and opportunities).
  - Section 5.4 (Water isotopes and end-member mixing models in glacierized catchment) was renamed and moved here from the previous Section 5.6 (Mixing model limitations). We moved also previous Figure 11, which is no Figure 10. We simplified the figure and removed the part showing results without transfer. The message was too complex and missed a bit the main focus of the paper. We also modified the text to focus more directly on the use of isotopes to separate snow and ice.
  - Section 5.5 (Water isotopes for hydrological routing in glacierized catchment) was moved and adapted from previous Section 5.3 (Limitations of the hydrological transfer model). We adapted the text to focus more on the use of isotopes to better inform models for the parametrization of the routing scheme. Figure 10 was moved here and renamed Figure 11.
  - The three last sections now address the three main questions of the introduction.
- Conclusion: we simplified the conclusion and removed the long final part, which is now summarized in 3 main bullet points answering the question in the introduction.

Specific comments

**Comment 1**

*Lines 50-53: These basic sentences about fractionation can be skipped because they are not meaningful for the introduction. Given the topic of the manuscript, if the authors want to define isotopic fractionation, I think they should provide an example regarding the snowpack instead of vapour masses and precipitation.*

**Answer 1**

Since the target readers of the paper include glacio-hydrological modelers who may not be fully familiar with isotopes, we believe a short theoretical background on isotopes is needed, including the reason why snowfall is more depleted in heavy isotopes than rain. We improved this with an example more linked to the snowpack.

**Changes** : We moved the theoretical information about isotopes to a new section 2.5 (Theoretical background on water stable isotopes) to better structure the manuscript. We moved a few paragraphs from the introduction and results here.

**Comment 2**

*Equation 1: The sentence at Lines 54-55 is enough and does not require Equation 1.*

**Answer 2**

We skipped the equation 1.

**Comment 3**

*Lines 288-289: Is there any consideration based on the uncertainty in the isotopic analysis or is it just a simple preference for δ2H?*

**Answer 3**

The analytical error for both d2H and d18O is very limited. However, slight evaporative fractionation may have occurred for samples acquired using the automatic sampler in the river, which samples were not protected against evaporation (using oil for instance). For those samples, evaporative fractionation would have influenced more the value of d18O than d2H. For this reason, too, d2H was retained.

**Changes** : we think this information is too detailed for the paper and no changes were provided here.

**Comment 4**

*Lines 443-444: 'likely due to the preferential elution of solutes in the snowpack (Costa et al., 2020)' belongs to the discussion.*

**Answer 4**

**Changes** : this sentence was removed from the manuscript.

**Comment(s) 5**

*Line 447: 'As suggested in other studies (Beria et al., 2018)…' belongs to the discussion.*

*Lines 450-452: This sentence also belongs to the discussion.*

*Lines 461-462: This sentence also belongs to the discussion.*

**Answer 5**

**Changes**: The two first paragraphs were moved to the new section 2.5. The last paragraph was moved to the discussion (section 5.5).

**Comment 6**

*Lines 497-499: This sentence also belongs to the discussion.*

**Answer 6**

**Changes**: moved to section 5.1.

**Comment 7**

*Section 5.2.1 and Figure C2: It is interesting to note that ice melt had a relatively small spatio-temporal variability, despite the larger variability observed in ice surface samples. This is in agreement with the isotopic composition of ice and meltwater samples collected over a glacier surface in the Italian Alps (Zuecco et al., 2019). In Figure C2, I wonder whether the first ice melt samples collected in July 2019 were affected by mixing with snowmelt or recent rain water.*

**Answer 7**

Thank you for the comment. In Figure C2, it is indeed possible that the ice melt samples on 12 July 2019 were contaminated by snow melt as they were not collected as far away from the snow line as other samples.

**Changes**: we added the reference and also added a sentence to direct the reader to figure C2.

**Comment 8**

*Section 5.6 and Conclusions: By reading these two sections, I wonder whether stables isotopes of hydrogen and oxygen represent a real added value for the model application and for improving our understanding of hydrological processes in glacierized catchments. It looks like that the huge effort and the still-present challenges make the application of isotopes not that appealing compared to other tracers (e.g., major ions, trace elements, other isotopes, artificial tracers) that could better help discriminating the end members in stream runoff.*

**Answer 8**

Yes. This is one of the messages and outcomes of the paper: water isotopes of snow in a glaciated catchment may vary strongly spatially and temporarily, but even more importantly, may completely overlap the ice composition, at least in temperate glaciers in the European Alps. As a result, snow isotopes applications should be considered with care and not be confused with ice in mixing models. They may still be useful to analyze the transfer of rain through such a system, as discussed in Figure 10 (now figure 11). We agree that other (conservative) tracers may be needed depending on the application.

**Changes** : we adapted the conclusion and the last two sections of the discussion to bring a clearer message in the line of the reader's comments. We do not discuss other tracers directly in the paper since it was not the direct goal of the paper.

**Comment 9**

*Lines 787-820: Given the length of the manuscript and of the conclusions (quite long), I suggest organizing this text using bullet points and reducing the paragraph starting at Line 799. Bullet points and a shorter text should help the reader to understand the novelty and the take home messages of this manuscript.*

**Answer 9**

**Changes** : Yes, we simplified and adapted the conclusion with bullet points.

Technical corrections

**Comments**

- *Line 145: 'Snow profiles for isotopic analysis' instead of 'isotopic snow profiles'. Please change the term at Line 147, as well.*
- *Line 245: It should be Walter et al. (2005) instead of Todd Walter et al. (2005).*
- *Line 265: It should be Walter et al. (2005).*
- *Line 288: Please remove 'water' before 'stable'.*
- *Line 334: Please replace 'remain below' with 'more depleted than'.*

**Answer**

We thank the reviewer for these technical corrections.

**Changes**: Changed according to reviewer comments except for "Todd Walter" which seems to be a composed surname.